

Title: A multi-model assessment of the early last deglaciation (PMIP4 LDv1): The
meltwater paradox reigns supreme

Corresponding author: Brooke Snoll[1]

Email for correspondence: ee19b2s@leeds.ac.uk

Co-authors: Ruza Ivanovic[1], Lauren Gregoire[1], Sam Sherriff-Tadano[1], Laurie Menviel[2], Takashi
Obase[3], Ayako Abe-Ouchi[3], Nathaelle Bouttes[4], Chengfei He[5], Feng He[6], Marie Kapsch[7], Uwe
Mikolajewicz[7], Juan Muglia[8], Paul Valdes[9]

Affiliations: [1]University of Leeds, [2]Climate Change Research Centre, University of New South Wales
Sydney, [3]University of Tokyo, [4]LSCE, [5]University of Miami, [6]Univeristy of Wisconsin, [7]Max Planck
Institute for Meteorology, [8]CESIMAR, [9]University of Bristol





## Abstract

Transient simulations of the last deglaciation have been increasingly performed to better understand the processes leading to both the overall deglacial climate trajectory as well as the centennial- to decadal- scale climate variations prevalent during deglaciations. The Paleoclimate Modelling Intercomparison Project (PMIP) has provided a framework for an internationally coordinated effort in simulating the last deglaciation (~20 – 11 ka BP) whilst encompassing a broad range of models. Here, we present a multi-model intercomparison of 17 simulations of the early part of the last deglaciation (~20 – 15 ka BP) from nine different climate models spanning a range of model complexities and uncertain boundary conditions/forcings.

A main contrasting element between the simulations is the method by which groups implement freshwater fluxes from the melting ice sheets and how this forcing then impacts ocean circulation and surface climate. We find that the choice of meltwater scenario heavily impacts the deglacial climate evolution, but the response of each model depends largely on the sensitivity of the model to the freshwater forcing as well as to other aspects of the experimental design (e.g., $CO_2$ forcing or ice sheet reconstruction). There is agreement throughout the ensemble that warming begins in the high latitudes associated with increasing insolation and delayed warming in the tropics aligned with the later increases in atmospheric $CO_2$ concentration. The delay in this warming in the tropics is dependent on the timescale of the $CO_2$ reconstruction used by the modelling group. Simulations with freshwater forcings greater than 0.1 Sverdrup (Sv) after 18 ka BP experience delayed warming in the North Atlantic, whereas simulations with smaller freshwater forcings begin deglaciating sooner. All simulations show a strong correlation between North Atlantic temperatures, atmospheric $CO_2$ concentrations, and the AMOC. In simulations with a freshwater forcing greater than 0.1 Sv, North Atlantic temperatures correlate strongly with changes in the AMOC. Simulations with a smaller freshwater forcing show stronger correlations with atmospheric $CO_2$. This indicates that the amount of meltwater strongly controls the climate trajectory of the deglaciation. Comparing multiple simulations run by the same model demonstrate model biases by showing similar surface climate spatial patterns despite the use of different ice sheet reconstructions and/or meltwater flux scenarios. Simulations run with different models, but similar boundary conditions, have provided insight into the sensitivity of individual models to particular forcings, such as the amount freshwater forcing, which has been highly debated in previous studies.

This debate has stemmed from the so-called 'meltwater paradox' that exists in choosing how much meltwater to input into simulations of the last deglaciation (i.e., large and geologically inconsistent meltwater forcings that successfully produce abrupt climate events versus



glaciologically realistic meltwater fluxes that do not). The results of this research highlight how
important this decision is.

## 1. Introduction

At the onset of the most recent deglaciation, ~19 thousand years before present (ka BP), ice sheets
that covered the Northern Hemisphere at the Last Glacial Maximum (LGM; Dyke 2004; Lambeck et al.
2014; Hughes et al. 2016) started to melt (Gregoire et al. 2012), Earth began to warm (Jouzel et al.
2007; Buizert et al. 2014), and sea levels rose (Lambeck et al. 2014). Known as the *last deglaciation*,
this time period is defined by major, long-term (order of ten thousand years) climate transitions from
the most recent cold glacial to the current warm interglacial state, as well as many short-term, decadal
to centennial-scale warmings and coolings of more than 5 °C (de Beaulieu and Reille 1992;
Severinghaus and Brook 1999; Lea et al. 2003; Buizert et al. 2014). These short-term abrupt
temperature changes are often also accompanied by sudden reorganisations of basin-wide ocean
circulations (e.g., Roberts et al. 2010; Ng et al. 2018) and jumps in sea level of tens of meters in a few
hundred years (e.g., Deschamps et al. 2012; Lambeck et al. 2014).

Abrupt climate changes observed in the early last deglaciation such as the Greenland cold
period known as Heinrich Stadial 1 (between ~18.5 and 14.7 ka BP; Broecker and Putnam 2012;
Huang et al. 2014, 2019; Crivellari et al. 2018; Ng et al. 2018) and the Bølling Warming (an abrupt
warming that occurs ~14.7 ka BP in Greenland at the end of Heinrich Stadial 1; Severinghaus and
Brook 1999; Lea et al. 2003; Buizert et al. 2014), are often attributed to changes in the Atlantic
meridional overturning circulation (AMOC). The strength and structure of the ocean circulation is a
key control on the North Atlantic and Arctic climate and is dependent on the stratification of the water
layers in crucial convection sites in the North Atlantic (Lynch-Stieglitz et al. 2007; McCarthy et al.
2017). When the AMOC is strong, more heat is transported towards the North Atlantic causing
regional warming in Greenland and the North Atlantic (Rahmstorf 2002).

There is debate on the strength of the LGM AMOC and how this initial state impacts the
preceding climate change of the deglaciation. Previous observations (e.g., Lynch-Stieglitz et al. 2007;
Böhm et al. 2015; Lynch-Stieglitz 2017) suggest a weaker and shallower AMOC than present-day
during the LGM, whilst more recent modelling studies show a deep and strong ocean circulation (e.g.,
Menviel et al. 2011; He et al. 2021; Sherriff-Tadano and Klockmann 2021; Kapsch et al. 2022; Snoll et
al. 2022) due to the presence of thick ice sheets (Oka et al. 2012; Sherriff-Tadano et al. 2018; Galbraith
and de Lavergne 2019). Multiple more recent data assimilation modelling studies (e.g., Menviel et al.
2012, 2017; Muglia and Schmittner 2021; Wilmes et al. 2021; Pöppelmeier et al. 2023b) have shown



agreement with previous observations by reproducing a weak and shallower LGM AMOC relative to the pre-industrial or Holocene. However, several ocean circulation proxy studies (e.g., McManus et al. 2004; Gherardi et al. 2005, 2009; Ivanovic et al. 2016; Ng et al. 2018), using the $^{231}Pa/^{230}Th$ ratio, demonstrate some consensus of a vigorous but shallower AMOC coming out of the LGM (relative to the modern day) that subsequently weakened and shallowed (but remained active; Bradtmiller et al. 2014; Repschläger et al. 2021; Pöppelmeier et al. 2023b) during the abrupt transition to Heinrich Stadial 1. Palaeoproxy records (e.g., McManus et al. 2004; Lynch-Stieglitz 2017; Ng et al. 2018) and modelling studies (e.g., Liu et al. 2009; Menviel et al. 2011; He et al. 2021) suggest that the ocean circulation strength remains weak until its rapid resumption at the end of the stadial, when the Bølling Warming occurs.

The AMOC pattern can be perturbed easily by several climate feedbacks. For example, if freshwater is deposited into the critical convection sites in the subpolar North Atlantic, i.e., the Labrador Sea and Nordic Seas, locations of high sensitivity to wind patterns and sea ice formation, the circulation strength can be disrupted (Rahmstorf 1999). Evidence from several sites report sea level rise, and therefore a freshwater flux, as early in the deglaciation as 19.5 ka BP, attributed to widespread retreat of Northern Hemisphere ice sheets in response to an increase in northern latitude summer insolation (Yokoyama et al. 2000; Clarke et al. 2009). Carlson and Clark (2012) concluded that the LGM was terminated by a rapid 5 – 10 meters sea level rise between 19.5 and 19 ka BP, and sea levels rose a further 8 – 20 meters from ~19 to 14.5 ka BP with the melt of the Laurentide and Eurasian ice sheets. More recent reconstructions of sea level and ice volume change suggest a similar view with ~10-15 meters of sea level rise between the end of the LGM (~21 – 20 ka BP) and 18 ka BP and an additional ~25 meters before 14.5 ka BP (Lambeck et al. 2014; Peltier et al. 2015; Roy and Peltier 2018; Gorbarenko et al. 2022). In some cases where meltwater fluxes are applied to the North Atlantic in model simulations, rapid decreases of up to 10 °C in temperature occur, resembling the transition to Heinrich Stadial 1 (e.g., Ganopolski and Rahmstorf 2001; Knutti et al. 2004; Brown and Galbraith 2016; Menviel et al. 2020).

Transient simulations of the last deglaciation have been increasingly performed to better understand the multi-millennial scale processes and the shorter and more dramatic climate changes by examining dynamic and threshold behaviours (Braconnot et al. 2012), determining the effects of temporally varying climate forcings, and identifying what mechanisms in the model can cause recorded climate signals (see section 1.2 by Ivanovic et al. (2016) and examples therein). In turn, these simulations also provide us with the opportunity to test the ability of models to simulate climate





processes and interactions as well as different hypotheses for drivers of change (i.e., climate triggers, interactions, and feedbacks).

One particularly challenging aspect in the experimental design of last deglaciation
simulations is prescribing ice sheet evolution and the resultant freshwater flux and sea level rise. Notwithstanding the qualitative rationale for why ocean-bound meltwater disrupts ocean circulation
and climate (McManus et al. 2004; Clarke et al. 2009; Thornalley et al. 2010), it has been recently argued that climate models are too sensitive to freshwater fluxes under some conditions. For example,
data reconstructions suggest only a small change in AMOC ~11.7 to 6 ka BP, whereas CCSM3 simulated a greater response to the freshwater forcing associated with the final Northern
Hemisphere deglaciation at this time (He and Clark 2022), when sea level rose by 50 meters during this interval (Lambeck et al. 2014; Cuzzone et al. 2016; Ullman et al. 2016). This result may be quite
model dependent, and we note that others had previously suggested the converse: that model responses to freshwater (and other) forcings could be too muted, from what we understand of past
climate change (Valdes 2011). Certainly, to disrupt climate in a Heinrich Stadial-like way, many previous glacial simulations have required quite large meltwater fluxes compared to what may be
inferred from geological records (Kageyama et al. 2013). This remains an interesting point of contention, and certainly some models no longer appear as 'stable' as they once did . Moreover, the
sensitivity of the North Atlantic Ocean circulation to glacial melting is poorly constrained.

There are, however, strong indications that the impact of oceanic freshwater fluxes is highly
dependent on the location that they enter the ocean (depth and latitude/longitude), as it determines the efficiency of convection disruption (e.g., Roche et al. 2007, 2010; Smith and Gregory 2009; Otto-
Bliesner and Brady 2010; Condron and Winsor 2012; Ivanovic et al. 2017; Romé et al. 2022). Similarly, the background climate and ocean state may also be important for how responsive ocean circulation
is to freshwater forcing–e.g., whether AMOC is already strong and deep or weak and shallow (Bitz et al. 2007; Schmittner and Lund 2014; Dome Fuji Ice Core Project Members: et al. 2017; Pöppelmeier
et al. 2023a), or specifically where deep water formation occurs (Smith and Gregory 2009; Roche et al. 2010). The choice of a model's boundary conditions in the palaeo setting (e.g., ice sheet geometry)
can influence its sensitivity to freshwater perturbation (e.g., compare the results of Romé et al. (2022) with those of Ivanovic et al. (2018)—Romé's low freshwater forced LGM simulations have an
oscillating AMOC, whereas Ivanovic's do not—different ice sheet reconstructions are used in both; and see Kapsch et al. (2022) who simulate the last deglaciation testing the climate response to
different ice sheet reconstructions). Ice sheet geometry specifically has been demonstrated to be linked with AMOC strength due to the effect of ice sheet height on surface winds and wind-driven





158 gyres, which can increase the northward transport of salty waters. Multiple model studies (e.g., Ullman et al. 2014; Löfverström and Lora 2017; Sherriff-Tadano et al. 2018; Kapsch et al. 2022) have

160 shown that a thicker Laurentide ice sheet results in a stronger AMOC. Hence, the influence of deglacial ice sheet meltwater on AMOC is likely highly dependent on both the model, choice of boundary

162 conditions and forcings, and the initial ocean condition.

   Previous modelling efforts (e.g., Liu et al. 2009; Roche et al. 2011; Menviel et al. 2011;

164 Gregoire et al. 2012) have performed transient simulations to learn more about the last deglaciation and the interaction between ocean and atmosphere. Liu et al. (2009) were the first to publish a

166 synchronously coupled atmosphere-ocean general circulation model simulation of the last deglaciation, henceforth referred to as *TraCE-21ka*. In this study, a freshwater flux was used to

168 regulate the AMOC to achieve a set of target ocean circulation, surface air temperature, and sea surface temperature conditions as interpreted from a selection of proxy records in multiple locations

170 between the LGM and the onset of the Bølling Warming (see Fig. 1 by Liu et al. (2009)), preceded by a switch to a geologic reconstruction of freshwater forcing (He 2011). Largely following the strategy

172 of *TraCE-21ka*, He et al. (2021) performed an updated version of the simulation with an isotope enabled model, referred to as *iTraCE*. Similarly, they also implemented a freshwater flux tuned to their

174 interpretation of an ocean circulation proxy record (McManus et al. 2004) between the LGM and the onset of the Bølling Warming and produced model output timeseries that very closely resemble water

176 isotope observations in Greenland (GISP2; Grootes et al. 1993), northern (Kulishu Cave; Ma et al. 2012) and southern China (Hulu Cave; Wang et al. 2001), and southern Asia (Mawmluh Cave; Dutt et

178 al. 2015). Menviel et al. (2011) adopted a similar approach, using freshwater fluxes tuned to ocean circulation and climate records (McManus et al. 2004 and Alley 2000 respectively) in order to better

180 understand abrupt changes in the Southern Hemisphere and their possible link to the so-called bi-polar seesaw (Broecker 1998; Stocker 1998) versus Antarctic melting.

182   The meltwater inputs used in each of these simulations, however, do not follow ice sheet reconstructions (e.g., see Ivanovic et al., 2018). Instead, the meltwater fluxes are, on occasion over

184 twice as large as suggested by ICE-6G_C VM5a (henceforth 'ICE-6G_C'; Argus et al. 2014; Peltier et al. 2015) and GLAC-1D (Tarasov and Peltier 2002; Tarasov et al. 2012; Briggs et al. 2014; Ivanovic et al.

186 2016). Furthermore, the freshwater flux must then be shut off to reinvigorate the AMOC and instigate the Bølling Warming, ending Heinrich Stadial 1, but this is at the same time as recorded rise in global

188 sea level of 12-22 meters in ~350 years or less, known as Meltwater Pulse 1a (Deschamps et al. 2012). This creates a meltwater paradox, where the freshwater forcing required by models to produce



recorded climate change is broadly in opposition to the meltwater history reconstructed from ice sheet and sea level records.

Simulations performed by Kapsch et al. (2022) and Snoll et al. (2022) add weight to this so-called meltwater paradox. They use meltwater forcing scenarios in accordance with observable ice

volume change but have not been able to replicate the AMOC or surface air temperature proxy records. Instead, the AMOC remains stronger than ocean circulation records suggest for Heinrich

Stadial 1, and the models simulate an abrupt *cooling* at ~14.5 ka BP instead of the Bølling *Warming*.

      The picture is further confounded by Gregoire et al. (2012), who forced an ice sheet model

with a transient climate simulation of the last deglaciation that included meltwater pulses, such as Meltwater Pulse 1a, but which did not experience major AMOC or surface temperature perturbations

as a result of the freshwater (e.g., see Fig. S2 by Gregoire et al. (2012)).

      Similar simulations of the last deglaciation have been run with no prescribed meltwater or a

202 meltwater forcing that is applied as a global salinity adjustment (i.e., rather than localised surface forcing). Without the use of the freshwater forcing, these simulations do not reproduce any abrupt

climate change events during the deglaciation. For instance, those described by Snoll et al. (2022) are similarly not able to replicate the AMOC or surface air temperature proxy records, but also do not

simulate the abrupt cooling at ~14.5 ka BP and instead, display a gradual warming throughout the deglaciation. Roche et al. (2011) performed a no-melt simulation of the last deglaciation that matches

closely with NGRIP temperatures (North Greenland Ice Core Project, 42.32° W, 75.01° N) until 16 ka BP when Heinrich Stadial 1 is observed. Yet, at ~11 ka BP, after the abrupt events occur, the simulation

and NGRIP temperatures meet each other again. Bouttes et al. (2023) also conduct a no-melt as well as meltwater forced simulation, but with an adaptation of the same model used by Roche et al. (2011).

Comparably, the new simulations also match closely with the NGRIP temperatures, but only until 17 ka BP, when they start to warm more rapidly and do not replicate Heinrich Stadial 1 cooling or the

subsequent Bølling Warming.

      The simulation performed by Obase and Abe-Ouchi (2019), is unique in that it is able to

216 simulate a weak AMOC during the onset of the deglaciation and the Bølling Warming without releasing (and then stopping) an unrealistically large amount of freshwater. Instead, they input a

218 gradually increasing amount of meltwater that remains at or below the level of ice volume loss in the reconstruction. At ~14.7 ka BP, the AMOC and Greenland surface air temperature abruptly increases

due to the gradual warming of the Southern Ocean and increased formation of Antarctic Bottom Water reaching a threshold whilst the meltwater flux in the North Atlantic is maintained, overcoming

the effect of the meltwater. This simulation still does not consider Meltwater Pulse 1a and has lower



than observed meltwater input before that point, yet it is distinctive in its ability to replicate a weak ocean circulation in the early deglaciation and the Bølling Warming even with a continuous freshwater flux.

Despite the decades of research simulating the last deglaciation and numerous observable records of this time period, uncertainty still remains about the mechanisms that cause the recorded climate signals as well as how to replicate them 'realistically' in model simulations, and therefore how to unravel the meltwater paradox. Furthermore, $CO_2$ and orbital forcing are also shown to impact the course of the deglaciation and the occurrence of abrupt climate changes (i.e., results shown by Oka et al. 2012; Klockmann et al. 2016, 2018; Zhang et al. 2017; Sherriff-Tadano et al. 2018), and also potentially modulate the sensitivity of the AMOC to freshwater fluxes (Obase and Abe-Ouchi 2019; Sun et al. 2022). Liu et al. (2009) demonstrate that the warming in *TraCE-21ka* between 17 and 14.67 ka BP is dominated by the $CO_2$ forcing (over the orbital forcing; see their Fig. S6a), which coincides with the first major rise of atmospheric $CO_2$ in their simulation. Whereas Gregoire et al. (2015) find, using a couple climate-ice sheet model (FAMOUS-GLIMMER) and following the deglacial simulation of Gregoire et al. (2012), that by 9 ka BP, orbital forcing caused 50% of the reduction in ice volume, greenhouse gases 30%, and the interaction between the two caused the last 20% in their simulations. Sun et al. (2022) shows the affect that these forcings have on the sensitivity of the AMOC with the use of multiple sensitivity experiments from an Earth system model (COSMOS). Their experiments demonstrate that a weak AMOC (in a Heinrich Stadial 1-like state, for example) is more likely to recover (like that of the Heinrich Stadial 1 to Bølling Warming transition) with a higher atmospheric $CO_2$ concentration, and that larger ice sheets result in a stronger AMOC that is less sensitive to meltwater fluxes. These findings highlight the importance of solving the convolved issue of model sensitivity to specific forcings/boundary conditions and the initial climate condition, and model dependency of simulation results—the crux of the remaining unknowns.

The Paleoclimate Modelling Intercomparison Project (PMIP) has been internationally coordinating multi-model simulations for over 30 years (Braconnot et al. 2007, 2012) to tackle such unknowns. The PMIP working groups have developed protocols to facilitate the effort of simulating past global climate changes by providing an outline of the model set-up to use (i.e., boundary conditions and forcings), allowing for easier cross-model comparison and the international pooling of resources to examine differing hypotheses to explain past climates, including uncertain model inputs (such as forcings and boundary conditions). As part of PMIP4, the last deglaciation protocol (Ivanovic et al. 2016) encompasses a broad range of models and is intentionally designed to be flexible, in order to be a suitable technical and computational undertaking for all participants and to



allow exploration of different climate scenarios. Instead of one specific and rigid configuration for the experiment design, modelling groups are given a choice of recommended forcings and boundary

conditions. Thus, analysing model output of multiple simulations of the last deglaciation provides the opportunity to look at differences between experimental designs and their impact on the onset of the

deglaciation using different models, though it can also be challenging. For example, it is difficult to compare directly different model results in a strict 'benchmarking' (Braconnot et al. 2012; Harrison

et al. 2015) framework, and there may be mismatched ranges of forcing within which to examine model sensitivities. However, to some extent this is already the case for transient simulations, even if

a strictly singular protocol is applied, as initial conditions or simulated events can impact later results, such as the starting AMOC condition or whether an abrupt cooling did or did not occur. Furthermore,

the technical implementation of boundary conditions/forcings in different models also leads to discrepancies in their configuration.

The Coupled Model Intercomparison Project (CMIP) has set a precedent for how to analyse ensembles of simulations with inconsistent experimental designs, such as those testing different

Shared Socioeconomic Pathways (SSPs), and the more recent incorporation of PMIP simulations in CMIP has allowed for a better understanding of model biases and therefore, how to better compare

their output. In addition, past flexible protocols and 'ensembles of opportunity' comprising simulations of contrasting model boundary conditions and forcings (e.g., Schmidt et al. 2012; Lunt et

al. 2012; Kageyama et al. 2013) have shown the value of experiment design flexibility for understanding climate change in a multi-model framework  without committing all resources to

specific paradigms. In short, it can be scientifically advantageous for groups to have the opportunity to determine their own focus whilst still contributing computational expensive and technically

challenging simulations towards an overarching, multi-model aim of learning more about physical behaviours in response to different simulated scenarios.

This study compares 17 simulations of the last deglaciation from nine different climate models with dissimilar experimental designs. Our aim is to take advantage of the numerous

simulations available to better understand the chain of events and mechanisms of climate changes in the early last deglaciation (i.e., from 20 to 15 ka BP), and our collective ability to simulate them. We

focus on the early deglaciation because although models may start differently from the LGM, the divergence from each other is smaller in comparison to further into the deglaciation. We investigate

the similarities and differences between the model results and what aspects of the variations in the model output can be attributed to the experimental design or model biases by analysing the transition

from the LGM, when and where the warming starts, and the impact of freshwater forcing. We also





address the meltwater paradox by discussing the results of meltwater scenario choices made by the

modelling groups.

## 2. Experiment designs across the ensemble

*Table 1: Detail of simulations referenced in the muti-model intercomparison.*

| Model | Resolution | Simulation Reference Name | Publication (model; simulation) | Simulation Duration (ka BP) | Prescribed Ice Sheet | GHG | Meltwater Scenario |
|---|---|---|---|---|---|---|---|
| CCSM3 | Atmosphere: 2.8° with 26 levels<br>Ocean: 1° x 0.3° with 25 levels | *TraCE-21ka* | Collins et al. 2006; Liu et al. 2009 | 22 – 0 | ICE-5G | Joos and Spahni 2008 | *TraCE-21ka* |
| FAMOUS | Atmosphere: 7.5° x 5° with 11 levels<br>Ocean: 3.75° x 2.5° with 20 levels | *FAMOUS* | Smith et al. 2008;Gregoire et al. 2012 | 20 – 13 | ICE-5G | Based on PMIP2; see Harrison et al. (2002) | *Bespoke (Fig. 1f)* |
| HadCM3B | Atmosphere: 3.75° x 2.5° with 19 levels<br>Ocean: 1.25° with 20 levels | *HadCM3_uniform* | Valdes et al. 2017; Snoll et al. 2022 and this study | 23 – 2 ka CE | ICE-6G_C | Loulergue et al. 2008; Schilt et al. 2010; Bereiter et al. 2015 | *Melt-uniform* |
|  |  | *HadCM3_routed* |  |  |  |  | *Melt-routed* |
|  |  | *HadCM3_TraCE* |  |  |  |  | *TraCE-like* |
| iCESM | Atmosphere: 2.5° x 1.9° with 30 levels<br>Ocean: 1° with 60 levels | *iTraCE* | Hurrell et al. 2013; He et al. 2021 | 21 – 11 | ICE-6G_C | Lüthi et al. 2008 | *TraCE-like* |
| iLOVECLIM | Atmosphere: 5.6° with 3 vertical levels<br>Ocean: 3° with 20 levels | *iLOVE_uniform_ice6gc* | Goosse et al. 2010; Bouttes et al. 2022 | 21 – 9 | ICE-6G_C | Loulergue et al. 2008; Schilt et al. 2010; Bereiter et al. 2015 | *Melt-uniform* |
|  |  | *iLOVE_routed_ice6gc* |  |  |  |  | *Melt-routed* |
|  |  | *iLOVE_uniform_glac* |  |  | GLAC-1D |  | *Melt-uniform* |
|  |  | *iLOVE_routed_glac* |  |  |  |  | *Melt-routed* |
| LOVECLIM | Atmosphere: 5.6° with 3 vertical levels<br>Ocean: 3° with 20 levels, dynamic vegetation model | *LOVECLIM* | Goosse et al. 2010; This study, but similar to simulations by Menviel et al. (2011) | 21 – 11.8 | ICE-5G | Köhler et al. 2017 | *TraCE-like* |
| MIROC | Atmosphere: 2.8° with 20 levels<br>Ocean: 1.4° with 43 levels | *MIROC* | Hasumi and Emori 2004; Obase and Abe-Ouchi 2019 | 21 – 13 | ICE-6G_C | Loulergue et al. 2008; Schilt et al. 2010; Bereiter et al. 2015 | *Bespoke (gradual increase)* |
| MPI-ESM-CR | Atmosphere: 3.75° with 31 levels<br>Ocean: 3° with 40 levels | *MPI_global_ice6gc* | Giorgetta et al. 2013; Kapsch et al. 2022 | 26 – 0 | ICE-6G_C | Köhler et al. 2017 | *Melt-uniform (Global meltwater flux)* |
|  |  | *MPI_routed_ice6gc* |  |  |  |  | *Melt-routed* |
|  |  | *MPI_routed_glac* |  |  | GLAC-1D |  | *Melt-routed* |
| UVic | Atmosphere: 3.6° x 1.8°<br>Ocean: 3.6° x 1.8° with 19 levels | *UVic_shorthosing* | Weaver et al. 2001; This study, but based on LGM simulations by Muglia and Schmittner (2015, 2021) | 21 – 14 | ICE-6G_C | dynamic | *Bespoke* |
|  |  | *UVic_longhosing* |  |  |  |  |  |

The comparison presented here is based on 17 simulations produced independently by eight

different palaeoclimate modelling groups, using nine different climate models (Table 1). Most groups

have followed the most recent PMIP4 last deglaciation protocol for their experimental design, while

others use older publications for boundary conditions or a more *bespoke* configuration depending on



their own modelling goals. The simulations from HadCM3, LOVECLIM, iLOVECLIM, iCESM, MIROC,
       and MPI modelling groups use greenhouse gas configurations on the AICC2012 age model of Veres et
al. (2013) (Fig. 1b). *FAMOUS* and *TraCE-21ka* use an older age model in which the deglacial rise in $CO_2$
       starts one thousand years later. All simulations prescribe insolation following Berger (1978). The
PMIP4 last deglaciation protocol recommends using the GLAC-1D (Ivanovic et al., 2016) and/or ICE-
       6G_C (Peltier et al., 2015) ice sheet reconstructions. HadCM3, iCESM, MIROC and UVic modelling
groups opted for ICE-6G_C, MPI and iLOVECLIM simulations use both ICE-6G_C and GLAC1-D, and
       *FAMOUS*, *LOVECLIM*, and *TraCE-21ka* use the older ICE-5G (Peltier 2004).

Freshwater forcing across the ensemble is more complex. The PMIP4 last deglaciation
       protocol recommends two different meltwater scenarios (*melt-routed* and *melt-uniform*) based on ice
volume change as calculated from the ice sheet reconstruction chosen by the modelling group (GLAC-
       1D and ICE-6G_C are recommended). The *melt-uniform* scenario is a globally uniform freshwater flux
or salinity adjustment through time applied throughout the whole ocean to conserve water mass
       during deglaciation of the ice sheets, whereas the *melt-routed* scenario is a distributed routing that
gives the flux of freshwater through time at individual meltwater river outlets along the coast
       (Ivanovic et al. 2016; Riddick et al. 2018 - used by MPI).

Because a large discrepancy between the simulations is the prescribed freshwater flux
       scenario (Fig. 1c-e), and ice sheet meltwater fluxes are known to have a major impact on ocean
circulation and climate (see above), the simulations have been grouped into four categories based on
       their meltwater forcing: *melt-routed*, *melt-uniform*, those based on the *TraCE-21ka A* simulation
(henceforth referred to as *'TraCE-like'*; Liu et al. 2009), and '*bespoke*' scenarios that fall outside of the
       other three categories. Within these categories, however, there is variation in how the freshwater
forcing is derived from the ice sheet reconstruction as well as in the technical implementation of the
       chosen meltwater scenario (for example, for the *melt-routed* and *melt-uniform* scenarios, see Wickert
2016, section 2.2.2 for HadCM3; Kapsch et al. 2022; section 2 for MPI; Bouttes et al. 2022, section 2.4
       for iLOVECLIM). For the *melt-routed* simulations, the modelling groups then release the calculated
meltwater flux to ocean grid cells according to the distribution calculated by the individual groups'
       drainage network models (see respective papers). For the *melt-uniform* simulations, HadCM3 and
iLOVECLIM modelling groups apply a globally uniform freshwater flux throughout the entire volume
       of the ocean, whereas the MPI modelling group applies a freshwater flux at the surface of the ocean
or land. Because of this nuance, the MPI *melt-uniform* simulation is instead labelled as a 'global
       surface meltwater flux' but is still placed in the *melt-uniform* category for our analysis.



We somewhat over-simplistically refer to PMIP4 meltwater scenarios as 'realistic', because they are based on the chosen ice sheet reconstruction prescribed in the simulation. Nonetheless, it is

important to note that the precise history of the meltwater flux (distribution and rates) remains quite uncertain, as hinted at by differences in the reconstructions. Between 20 and 15 ka BP, the 'realistic'

freshwater flux according to ICE-6G_C does not exceed 0.1 Sv and that according to GLAC-1D only exceeds 0.1 Sv as it nears Meltwater Pulse 1a.

In the *TraCE-like* simulations, the strategy of prescribing freshwater to induce an inferred AMOC history requires the freshwater flux to reach nearly 0.2 Sv or greater—twice the 'realistic'

amount based on sea level records (Fig. 1d; Carlson and Clark 2012; Lambeck et al. 2014).

For the *bespoke-freshwater* cluster of simulations, *MIROC* implements a gradually increasing

flux that always remains below the 'realistic' values. *FAMOUS* uses a reconstructed flux based on an earlier estimate from sea level records (produced as part of the ORMEN project; more information

provided by Gregoire (2010)), which follows the more up-to-date ice sheet reconstructions relatively closely except when a larger freshwater flux is applied at two points during Heinrich Stadial 1

(between 19 and 17 ka BP; corresponding to the acceleration of Northern Hemisphere ice loss, as noted by Carlson and Clarke, 2012, and the melt of the Eurasian ice sheet as reconstructed by Hughes

et al. (2016)). The UVic simulations use a total freshwater flux calculated as three times the sea level changes reconstructed by Lambeck et al. (2014); one scenario where the freshwater flux is applied

between 19 and 15 ka BP (*Uvic_longhosing*) and one where the flux is only applied between 19 and 17 ka BP (*Uvic_shorthosing*; Table 1).

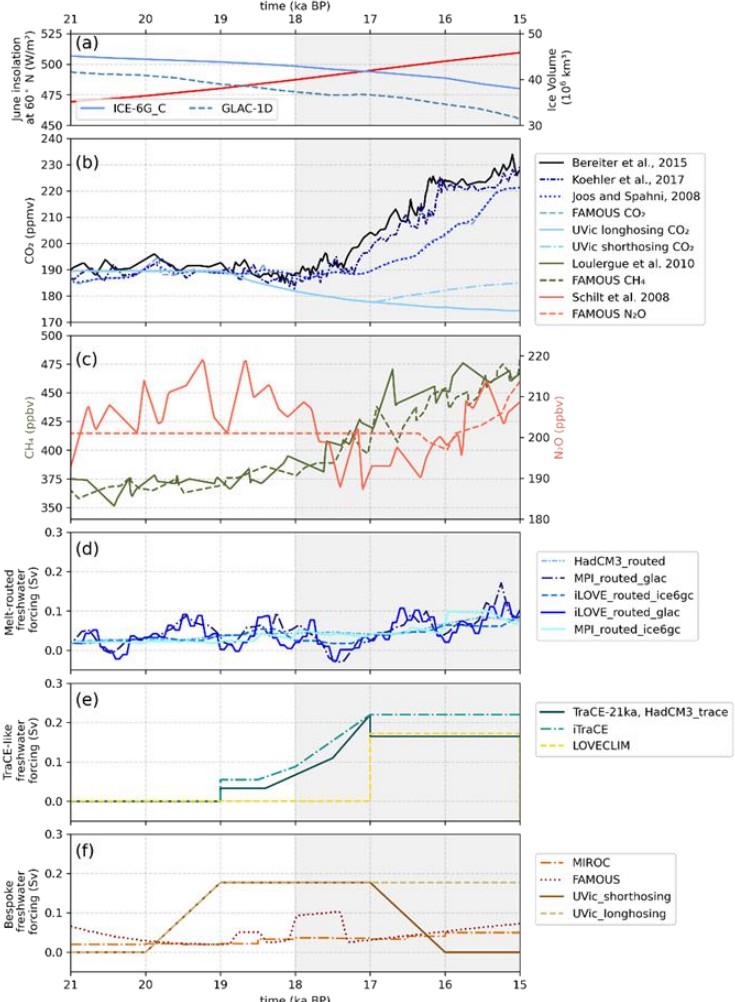

*Fig. 1: Climate forcings for the simulations. (a) Ice volume loss since the Last Glacial Maximum (LGM; 21 ka BP) as part of the*

352 *ICE-6G_C ice sheet reconstruction (Argus et al., 2014; Peltier et al., 2015) and the GLAC-1D ice sheet reconstruction (Tarasov*

*and Peltier 2002; Tarasov et al. 2012; Briggs et al. 2014; Ivanovic et al. 2016). (b) Atmospheric greenhouse gas*

*concentrations dependent on simulation set-up. (d)-(f) Freshwater flux (Sv) for simulations with imposed meltwater. Melt-*

*uniform simulations have the same total meltwater flux into the global ocean as melt-routed simulations (d), but in melt-*

356 *uniform scenarios, the freshwater is spread through the entire ocean or across the whole ocean surface (see main text) rather*

*than at point sources and hence are so diluted/uniformly distributed as to have limited direct forcing power.*

The UVic simulations include a dynamic carbon cycle model with prognostic atmospheric $CO_2$

aiming to replicate the sedimentary records of deep ocean carbon. The freshwater flux is, therefore,

tuned to replicate the AMOC structure associated with these sedimentary records, but the location of



the meltwater input is based on plume positions like those of the HadCM3 simulations. The UVic
       simulations are included in the broader comparisons presented here (i.e., Fig. 3Fig. 5). However,
because of their uniqueness of experiment design and motivation, the differences between the UVic
       simulations and the wider multi-model ensemble are too great for a more detailed comparison of
results, and they are therefore omitted from further analysis and discussion in this study.

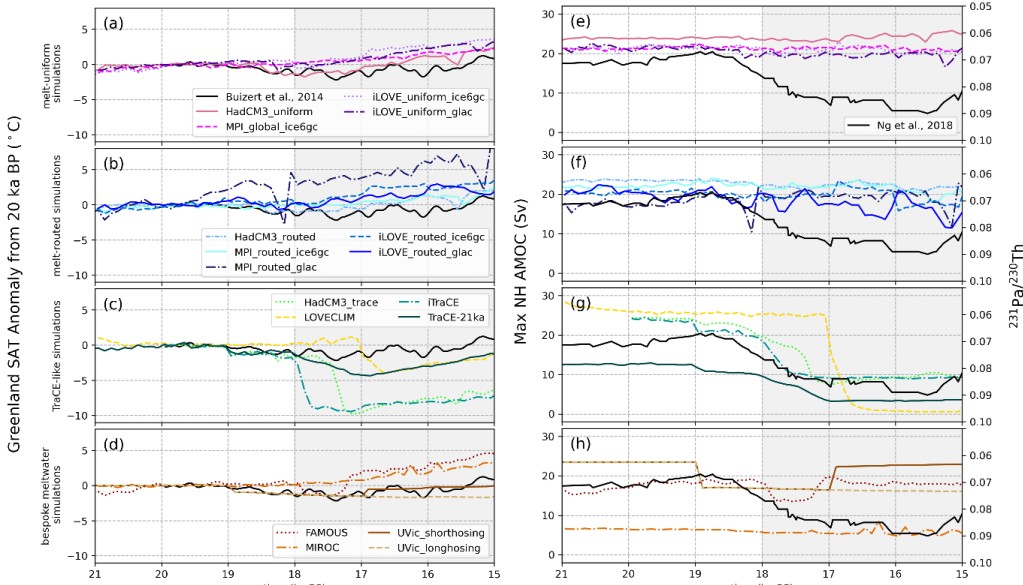

*Fig. 2: Centennial means for (a)-(d) Greenland (between 65 and 82° N and 30 and 55 °W) surface air temperature anomaly*
*from approximately the LGM (20 – 19.5 ka BP) for each simulation; (e)-(h) Maximum AMOC of the Northern Hemisphere at*
       *depth between 500 and 3500 meters. For comparison, (a)-(d) includes Greenland surface air temperature proxy record from*
*Buizert et al. (2014), plotted as an anomaly from 20 ka BP in black and (e)-(h) includes the AMOC proxy $^{231}Pa/^{230Th}$*
       *composite record published by Ng et al. (2018) in black - note arbitrary y-axis scaling.*



## 3. Results/Discussion

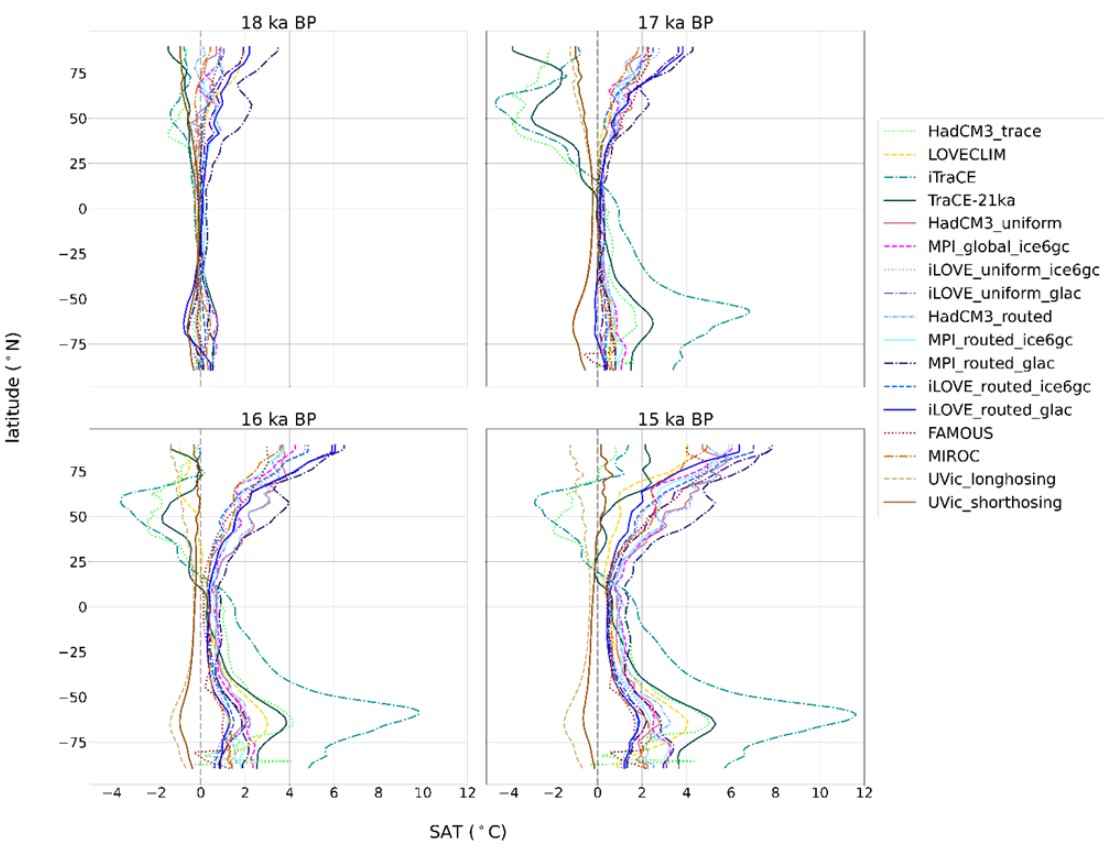

*Fig. 3: Zonal average of decadal mean surface air temperature as anomalies from the LGM (20 – 19.5 ka BP) for each*
*simulation. 18, 17, 16, and 15 ka BP are calculated as 60-year decadal means centred around the respective time period (e.g.,*
*from 17.97 to 18.03 ka BP for 18 ka BP).*

Here, we focus on the course of the deglaciation, how it is impacted by the freshwater forcing, and
how this relationship differs on a model-to-model and experimental design-to-experimental design
basis. The trajectory of the AMOC in the Northern Hemisphere for each simulation follows closely the
meltwater scenario chosen by the modelling group (Fig. 2). All the melt-routed, melt-uniform, and
bespoke freshwater scenarios display a similar pattern throughout the deglaciation with a gradual
warming of surface air temperature in the high latitudes and stronger warming compared to the
TraCE-like simulations in the Northern Hemisphere (Fig. 3). The similarity between the simulations
increases further into the deglaciation, with warming from the LGM in all regions by 16 ka BP for all
the melt-routed, melt-uniform, and bespoke freshwater scenarios (Fig. 3 and S1). The TraCE-like
simulations, however, do not follow the same trajectory, and the Northern Hemisphere, specifically



the North Atlantic, remains colder than at the LGM for most of the early deglaciation, with only

LOVECLIM and TraCE-21ka warming beyond the LGM in the North Atlantic by 15 ka BP (Fig. S2). This

colder region in the North Atlantic is evident in a multi-model mean of the ensemble where, on

average, the North Atlantic remains the coldest region throughout the early deglaciation (Fig. 4).

Around the onset of Heinrich Stadial 1 (18 ka BP), more discrepancy between simulations arises (as

indicated by disagreement even in the sign of change; Fig. 4) due to differences across the ensemble

in when and where the deglaciation begins as well as the freshwater fluxes applied.  However, by 15

398    ka BP, at least 70% of simulations agree with the sign of the mean in most areas. More disagreement

remains in the North Atlantic, the region of highest variance across the ensemble and where the

different freshwater fluxes used in the simulations have the most direct impact. The ensemble-wide

consensus of a warming climate, however, is consistent with the increases in North Hemisphere

summer solar insolation and atmospheric $CO_2$ (Fig. 1a, b).

*4.1 Timing of the deglaciation*

Between 20 and 15 ka BP, each of the meltwater groups, except for the *TraCE-like* simulations, have

relatively constant AMOC strengths. The *melt-uniform* simulations show the least millennial-scale

variability in AMOC *melt-uniform* (Fig. 2e). The *melt-routed* simulations, in comparison, have more

variation, aligned with the respective freshwater fluxes, and show a weakening trend starting at

~16.5 ka BP as freshwater input increases towards Meltwater Pulse 1a (Fig. 2f; Meltwater Pulse 1a

at 14.7 ka BP not shown). Like the *melt-routed* simulations, the *bespoke* simulations have more change

that is consistent with the freshwater flux, but for all *bespoke* simulations except for *UVic_longhosing*,

the AMOC strengths at 21 ka BP and at 15 ka BP are very similar.

The subset of *TraCE-like* simulations, on the other hand, show an abrupt weakening in AMOC

strength and an associated decrease in Greenland surface air temperature (anomaly from LGM,

calculated as anomalies from the 500-year time window from 20 – 19.5 ka BP) beginning between 18

and 17 ka BP depending on the simulation (Fig. 2c, g). The differences in timing of the decrease in

temperature for the *TraCE-like* simulations are likely associated with the differences in timing and

magnitude of the freshwater flux. For instance, *iTraCE* shows an earlier and more abrupt cooling than

*TraCE-21ka*. Despite both simulations reaching the same magnitude of freshwater at 17 ka BP, the

rate of freshwater input into the simulation between 19 ka BP and 17 ka BP differs. At 19 ka BP, there

is a larger increase in the freshwater flux in *iTraCE,* which corresponds to a smaller, but rapid decrease

in the AMOC strength and Greenland surface air temperature at this same time. After 19 ka BP, the

freshwater flux in *iTraCE* remains higher than in *TraCE-21ka*, and this is consistent with the sharper





decrease in surface air temperature in *iTraCE* in comparison to the relatively steady decrease in
temperature in *TraCE-21ka*.

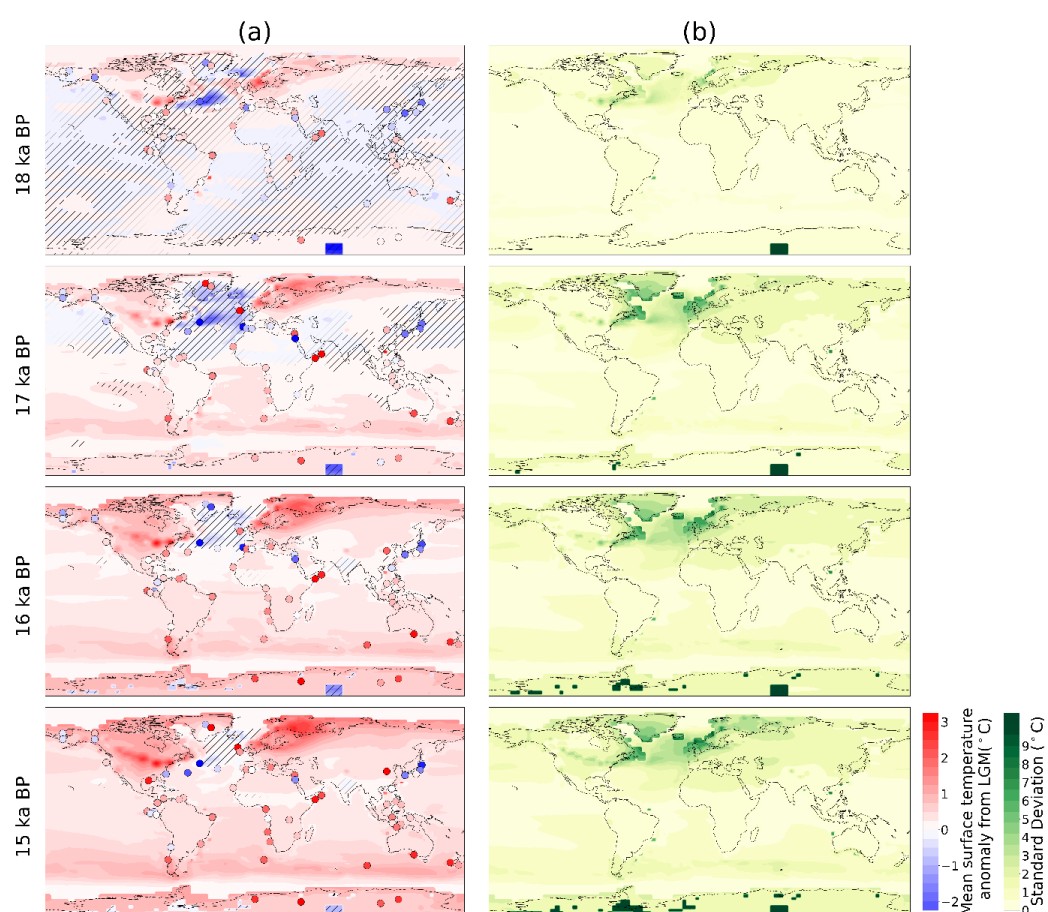

*Fig. 4: Column (a) Multi-model mean of decadal surface air temperature anomaly from the LGM (20 – 19.5 ka BP) at each
timestep labelled (not including the UVic simulations). Hatching denotes areas in which less than 70% of the models agree*
*with the sign of the mean. Column (b) Same as Column (a) but showing the variance. Filled circles show the proxy surface
temperature stack from Shakun et al. (2012) on the same colour scale. 18, 17, 16, and 15 ka BP are calculated as 60-year*
*decadal means centred around the respective time period (e.g., from 17.97 to 18.03 ka BP for 18 ka BP).*

*HadCM3_TraCE* uses the same meltwater scenario as *TraCE-21ka*, but instead of a gradual
response, there is a more abrupt decrease in the Greenland surface air temperature at ~17.5 ka BP
and temperatures drop. The drop is as low as in *iTraCE* (with respect to the LGM) and occurs after
the freshwater flux has decreased for both *TraCE-21ka* and *HadCM3_TraCE*. Note, that *TraCE-21ka*



and *HadCM3_TraCE*, however, are configured with different boundary conditions (i.e., *HadCM3_TraCE* uses greenhouse gas conditions on the AICC2012 timescale and the ICE-6G_C ice sheet

reconstruction, whereas the CCSM3 *TraCE-21ka* simulation uses ICE-5G) with the exclusion of the freshwater forcing. This suggests that the freshwater forcing is a dominant driver of the abrupt

changes displayed in both simulations; however, the differences between them might contribute to the differences in sensitivity to the meltwater flux.

In addition, although the meltwater scenario for *LOVECLIM* is based upon *TraCE-21ka*, the freshwater flux begins later, at 17 ka BP. Presumably because of this, the decrease in surface air

temperature and AMOC strength is also delayed until 17 ka BP. The freshwater input is also much more abrupt in comparison to *TraCE-21ka* and *iTraCE*, corresponding to the rapid transition in the

AMOC and surface air temperature at 17 ka BP. The implications of these differences amongst the simulations in the *TraCE-like* meltwater group are further described in section 4.4.

The GLAC-1D ice sheet reconstruction has more variable meltwater input in comparison to ICE-6G_C, at least partly due to the more frequent updates of the ice sheet geometry and associated

boundary conditions (every 100 years compared to every 500 years; Fig. 1c). This more variable meltwater forcing presents itself in the higher variability of the AMOC strength and Greenland surface

air temperature (Fig. 2b, f; e.g., the sharp decline and subsequent increase in temperature and AMOC strength at ~18.5 ka BP in *MPI_routed_glac* that occurs at the same time as an increase in meltwater

release).

     All the simulations that do not follow the *TraCE-like* meltwater forcing follow a similar

trajectory throughout the deglaciation with a gradual warming of surface air temperature in Greenland, except for the UVic simulations. The UVic simulations differ presumably because of the

*bespoke* freshwater flux that ends earlier than the end of Heinrich Stadial 1 for the short-hosed simulation and after Meltwater Pulse 1a for the long-hosed simulation. The resultant impacts on the

dynamically simulated carbon cycle causes atmospheric $CO_2$ concentrations to decrease during AMOC weakening, which contradicts reconstructions of this time period (e.g., Bereiter et al. 2015; Ng et al.

2018). Hence, in *UVic_longhosing,* decadal surface air temperature remains cold throughout the onset of the deglaciation, and *UVic_shorthosing* does not begin to warm in the Northern Hemisphere until

the freshwater hosing is turned off at 17 ka BP (Fig. 2).

     Across the ensemble, significant warming (defined by using a one-sided Student t-test of 99%

confidence with 100-year samples of surface air temperature annual means and 20 – 19.5 ka BP as the LGM reference period) from the LGM occurs in most parts of the world by 18 ka BP except for in

the tropics where significant warming does not occur until as late as 15 – 16 ka BP (Fig. 5). The earlier





warming in the high northern latitudes is likely  associated with the increase in insolation (Fig. 1a;

CAPE-Last Interglacial Project Members 2006; Park et al. 2019; Kapsch et al. 2021) and the impact of polar amplification; whereas the warming in the tropics is more delayed and correlates with the

timing of $CO_2$ concentration increases (Fig. 2Fig. 1b, and S3a-d). This pattern is aligned with the results from Roche et al. (2011) (see Fig. 4 by Roche et al. (2011); of which the analysis of Fig. 5 in

this study is based, but the LGM reference period is earlier than used here), that similarly show an earlier warming in the northern and southern high latitudes and delayed warming in the tropics. The

effect of the freshwater forcing on the global temperature, however, would not be incorporated in the no-melt simulations from Roche et al. (2011). Nevertheless, in the TraCE-like simulations, the

meltwater impact is evident by the strong cold anomalies in the North Atlantic, the region where most of the freshwater forcing is applied or drained into (Fig. 3 and 4). Warming in this region,

therefore, does not occur until much later in the deglaciation in comparison to the other simulations.

        This dissimilarity in the timing of the onset of the deglacial warming is also evident in global

surface air temperature anomalies from the LGM (Fig. 4 and S1). Early in the deglaciation, at 18 ka BP, there is disagreement between simulations as to the timing and magnitude of the warming as well

as to which regions. For instance, *MPI_routed_glac* has warmed ~4 °C in the North Atlantic by 18 ka BP, whereas *MIROC* still has colder regions throughout the tropics and Pacific with respect to the LGM

and has only started to warm in the high latitudes, most likely associated with insolation increases (Fig. S1).

By 16 ka BP, all the non-*TraCE-like* simulations show consistent warming throughout the globe, whereas the *TraCE-like* simulations still have the strong cooling in the North Atlantic associated

with the freshwater input (Fig. 3 and S1). In the *TraCE-like* simulations (most evident in *HadCM3_TraCE* and *TraCE-21ka*), the earlier deglacial warming in the Southern Hemisphere and the

delayed warming in the Northern Hemisphere are due to the bipolar seesaw (Broecker 1998; Stocker 1998) associated with the simulated slowdown of AMOC within Heinrich Stadial 1 (He et al. 2013),

with many areas not departing from their glacial climates until after 15 ka BP (Fig. 5). This is less evident in *LOVECLIM*, potentially because the cooling from the freshwater flux occurs later, at 17 ka

BP, and therefore, significant warming has already occurred beforehand (as also evident by the zonal surface air temperature means; Fig. 3).

Unlike the other non-*TraCE-like* simulations, *FAMOUS* does not warm in the tropics until later in the deglaciation due to its late rise in atmospheric $CO_2$ forcing. *TraCE-21ka* similarly warms later

further south in the tropics, compared to *HadCM3_TraCE* and *iTraCE*, which use the AICC2012 age model for greenhouse gas forcing. The iLOVECLIM and MPI simulations, in comparison, have



significant warming in most areas from the immediate onset of the deglaciation. Additional
       similarities are evident even amongst simulations that use the same model but different meltwater-
scenarios, e.g., between *HadCM3_uniform* and *HadCM3_routed* and between *MPI_routed_ice6gc* and
       *MPI_global_ice6gc.* The HadCM3 simulations have a matching cooling region around the Labrador Sea
and Gulf Stream, and the MPI simulations have a matching cooling region in the Nordic Seas that each
       persist until ~16 ka BP (more detail in section 4.3). UVic remains unique amongst the simulations
assessed in this study, because between 20 and 15 ka BP, most regions do not warm from the LGM.
       The $CO_2$ increase begins to take precedent in *UVic_shorthosing* after 17 ka BP and the melting ice
sheets in North America and Fennoscandia show familiar warming patterns in the Northern
       Hemisphere for ICE-6G_C: a pattern also evident in the other simulations using ICE-6G_C. This
warming still exists for *UVic_longhosing*, but with lower amplitude.





*Fig. 5: Year of first significant warming from 20 ka BP, where 'significant warming' is determined using a one-sided Student t-test with 100-year samples of surface air temperature annual means, 99% confidence, and 20 – 19.5 ka BP as the reference period (LGM). Hatching denotes where significant warming did not occur before 13 ka BP.*





Despite the disagreements with the timing of the deglaciation on an individual scale, the sign

of the multi-model mean of decadal surface temperature shares close agreement with the surface
temperature stack produced by Shakun et al. (2012), most significantly in the Southern Hemisphere

(Fig. 4). The median point-by-point difference between the multi-model mean and the proxy data is
less than 1 °C between 18 and 15 ka BP, with a median of only 0.015 °C at 18 ka BP that increases to

0.993 °C by 15 ka BP, indicating that the multi-model mean of the ensemble replicates the Shakun et
al. (2012) proxy stack relatively well, but that disagreement with the proxy record grows further into

the deglaciation. The largest discrepancies between the model output and reconstruction occur in the
North Atlantic and Greenland (after 18 ka BP), which are also areas of more disagreement across the

model ensemble (Fig. S4). This is the region where there are the most proxy records, and therefore
potentially the location in which the deglacial climate evolution is the best constrained (at least

compared to the Pacific sector, for example). The North Atlantic is also the region where most models
would show agreement for similar AMOC change, however these simulations show various AMOC

evolutions. The multi-model mean tends to be cooler than the proxy data in most locations, especially
the Southern Hemisphere, but is warmer in some parts of the North Atlantic, Alaska, and off the coast

of Japan. With much of the Northern Hemisphere too warm and Southern Hemisphere too cold, this
could argue that the representation of the bipolar seesaw is too pronounced in some simulations.

Interestingly, although the TraCE-like meltwater group represents the cold areas of the North Atlantic
well, those simulations have difficulty replicating the warmer core locations in this same region.

Conversely, the other meltwater groups present the opposite difficulty—they are better at replicating
the warmer regions of the North Atlantic while failing to represent the cold ones (not shown). This

suggests the potential need for subsequent investigations of broader model structure and how we
interpret reconstructions (i.e., specific data points).

### 4.2 Linking surface climate, ocean circulation, and greenhouse gas forcing

In every simulation, there is the expected interrelation between surface air temperature in the North
Atlantic, $CO_2$ concentration, and AMOC. As $CO_2$ increases, surface air temperature increases, as

demonstrated by the increasing trends on each panel of Fig. 6 surface air temperature is higher when
the AMOC is stronger, clearly shown by *LOVECLIM*. This relationship is illustrated further by the $R^2$

values determined by a linear regression model across the entire period between 20 and 15 ka BP on
a decadal temporal scale (Fig. 7Fig. 8). The results from the linear regression show that during the

period of 20 to 15 ka BP, surface air temperature in the *TraCE-like* simulations has a stronger positive
correlation with AMOC, and the other simulations in the ensemble have a stronger positive





correlation with $CO_2$. For instance, the *TraCE-like* simulations have higher $R^2$ values in the North

        Atlantic than the other meltwater groups, presumably because changes between AMOC and surface

air temperature correspond in the *TraCE-like* simulations between 20 and 15 ka BP, whereas the other

        simulations have a stable ocean circulation and very little temperature change during this time period

(Fig. 2). FAMOUS, which has a stronger freshwater forcing between 20 and 15 ka BP in comparison

        to the other non-*TraCE-like* simulations, also has higher $R^2$ values in the North Atlantic region, though

dampened relative to that of the *TraCE-like* simulation. The simulations with little AMOC and surface

        air temperature change show very low correlations between the two variables throughout the globe

(e.g., iLOVECLIM simulations, the ICE-6G_C MPI simulations, and MIROC). However, the *melt-routed*

        GLAC-1D simulations, in comparison to their ICE-6G_C same-model counterparts, exhibit higher

correlations, especially in the case of *MPI_routed_glac*.

                The positive slope in the North Atlantic region for the *TraCE-like* simulations demonstrates

the positive correlation between AMOC and surface air temperature changes, whereas the rest of the

        globe has a more negative correlation in most simulations, regardless of their meltwater group. This

relationship is representative of the bipolar seesaw. The *TraCE-21ka* simulation most clearly exhibits

        this bipolar connection between the Northern and Southern Hemispheres with a strong positive

correlation between AMOC and surface air temperature in the North Atlantic and a strong negative

        correlation in the Southern Ocean.





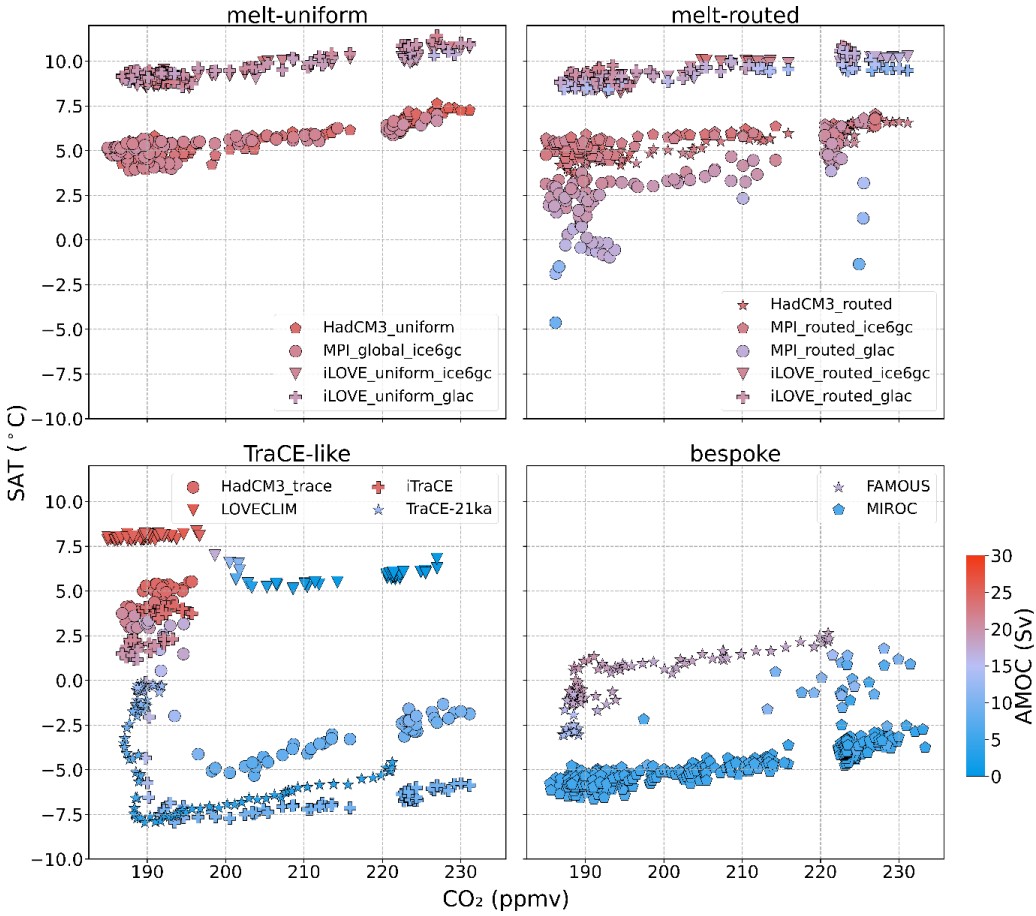

*Fig. 6: Absolute surface air temperature over the North Atlantic (between 35 and 60° N and -60 and 0° E) as a function of $CO_2$ concentration with symbols' shading representing the strength of the AMOC (Sv) split into groups defined by meltwater scenario. 50-year means are shown for each simulation except for MIROC, for which decadal means are shown to capture its temporally finer-scale variability. See Fig. S5 for the same analysis displayed as anomalies from 20 ka BP.*

The relationship between $CO_2$ and surface air temperature (Fig. 8) in the Northern Hemisphere is nearly opposite to the relationship between AMOC strength and surface air temperature for *HadCM3_TraCE*, *iTraCE*, and *TraCE-21ka,* with the areas of strong and positive correlation between AMOC and surface air temperature showing weaker and negative correlation between $CO_2$ and surface air temperature. This suggests that in the early deglaciation, if the AMOC is weakening/already weak because of the freshwater forcing when $CO_2$ starts to rise, the impact of $CO_2$ might be dampened or postponed in the Northern Hemisphere, whereas a strong correlation with surface air temperature remains in the Southern Hemisphere. The results by Sun et al. (2022) show



that the $CO_2$ increase could assist with the later transition out the weak AMOC state and into the
Bølling Warming (not shown here). Unlike the other simulations in the *TraCE-like* meltwater group,
the strength of correlation between AMOC and surface air temperature versus with $CO_2$ does not
change for *LOVECLIM*, potentially because the AMOC weakening and the $CO_2$ increase are concurrent.
The simulations with weaker correlation between $CO_2$ and surface air temperature in regions of the
tropics (e.g., *FAMOUS* and parts of Sub-Saharan Africa in *MIROC*, *MPI_global_ice6gc*, and
*HadCM3_routed*) also display delayed warming in these same locations (Fig. 5). Increases in obliquity
are shown to delay warming in the tropics, specifically in these same parts of Africa as well as India,
potentially due to increased cloud coverage and therefore cooling (Erb et al. 2013). In addition, the
lag between the start of the $CO_2$ concentration increase (~18 ka BP or later depending on the
timescale used) and the insolation increase (~20 ka BP) can disrupt the correlation between $CO_2$ and
surface air temperature and create a localised delay in warming of the tropics (as also demonstrated
in Fig. 5). Note that the analysis in Figures 7 and 8 only goes until 15 ka BP whereas the analysis in
Fig. 5 reaches until 13 ka BPFig. 5. The simulations with the very weak correlations between AMOC
and surface air temperature (iLOVECLIM, MPI simulations, and MIROC) demonstrate globally high
correlations with $CO_2$ except for a few concentrated regions. These regions of lower correlation are
similar between simulations run by the same model and could indicate changes in upwelling strength
during this time period.

It is important to note, however, that during the chosen time period, only the *TraCE-like*
simulations have strong and corresponding changes in the AMOC and surface air temperature. The
suggested relationships could be checked by continuing this study through the later parts of the
deglaciation to encompass greater amplitudes of change in the non-*TraCE-like* simulations.





*Fig. 7: Spatial correlation of AMOC strength and surface air temperature using a linear regression model for the time period*
       *20 - 15 ka BP using decadal means. Columns (a) and (c): R² values as a result of the linear regression. Columns (b) and (d):*
*corresponding slopes to simulation in Column (a) or (c) as a result of the linear regression.*





*Fig. 8: Spatial correlation of $CO_2$ concentration and surface air temperature using a linear regression model for the time*

*period 20 - 15 ka BP using decadal means. Columns (a) and (c): $R^2$ values as a result of the linear regression. Columns (b) and*

*(d): corresponding slopes to simulation in Column (a) or (c) as a result of the linear regression.*



### 4.3 Impact of different climate forcings on model output

In this study, we have included multiple simulations from the HadCM3, MPI, and iLOVECLIM modelling groups. These three modelling groups tested different PMIP4 boundary condition/forcing

options: for example, implementing the *melt-routed* or *melt-uniform* scenario for the same ice sheet and/or using different ice sheets and associated meltwater scenarios (Table 1). Experimenting with

the range of options the PMIP4 protocol enables us to review the impact of different climate forcings on the resultant model output.

The AMOC for each of the HadCM3, MPI, and iLOVECLIM simulations is impacted by the chosen meltwater scenario during the deglaciation (see section 4.1); however, between 21 and 15 ka

BP, the differences between the AMOC trajectory appear to be less affected by the meltwater scenario and instead more significantly affected by the choice of ice sheet reconstruction (Fig. 9). For instance,

when we compare the simulations with the different meltwater scenarios, but with the same ice sheet reconstruction (e.g., ICE-6G_C), i.e., *HadCM3_uniform* and *HadCM3_routed*, *iLOVE_uniform_ice6gc* and

*iLOVE_routed_ice6gc*, and *MPI_global_ice6gc* and *MPI_routed_ice6gc*, we notice multiple similarities between the deglaciation trajectory, spatially and temporally. For instance, the HadCM3 simulations

begin at a very similar surface air temperature in the North Atlantic at the start of the deglaciation (~4 °C at 21 ka BP) and follow a comparable warming trajectory until 15 ka BP (reaching ~7 °C; Fig.

9) despite the application of different meltwater scenarios, though the *melt-routed* simulation does remain colder in the North Atlantic than the *melt-uniform* simulation throughout the time period. In

addition, spatially, as anomalies from the LGM (Fig. 3 and S1), the simulations look almost indistinguishable. Both display surface air temperature cooling along the Gulf Stream, and warming

in locations of ice sheet melt, such as the Eurasian ice sheet in Fennoscandia and at the edge of the Laurentide ice sheet in North America. The most evident difference between the simulations is that

*HadCM3_uniform* is colder than *HadCM3_routed* in the Labrador Sea and warmer in the Norwegian Seas, corresponding with differences in sea ice concentration—*HadCM3_uniform* has a higher sea ice

concentration in the Labrador Sea than *HadCM3_routed* and a lower concentration in the Norwegian Seas (Fig. S6A and B). This pattern also corresponds to the dissimilarities in the convection sites

between the two simulations as the *melt-uniform* simulation has more convection further south, along the sea ice edge, and in the Norwegian Seas, whereas the mixed-layer depth in the *melt-routed*

simulation is deeper in the Labrador Sea (Fig. S6C). *HadCM3_TraCE* has the same dipole pattern as the other HadCM3 simulations, with cooling along the Gulf Stream and into Greenland and the

Labrador Sea, and warming over Fennoscandia; however, this signal is weak compared to the strong cooling in the North Atlantic due to the larger freshwater forcing applied.





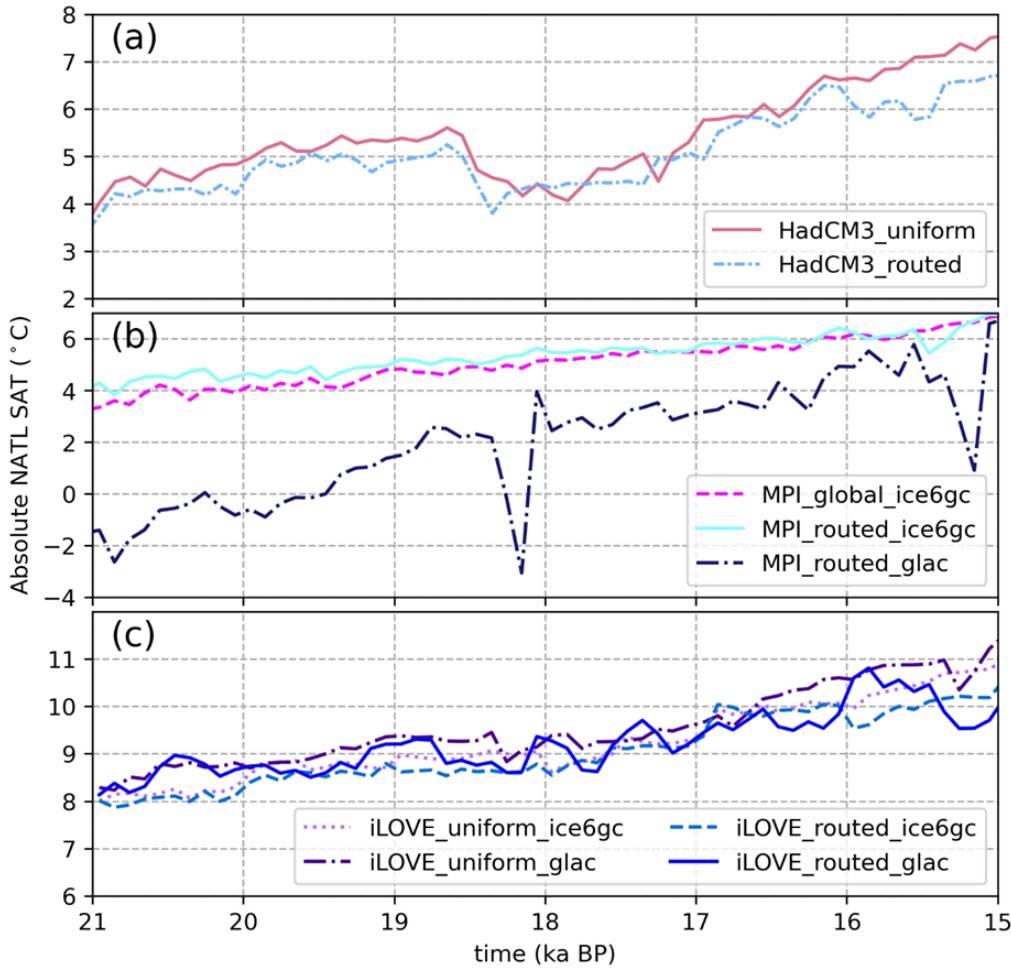

*Fig. 9: Absolute surface air temperature in the North Atlantic (between 35 and 60° N and -60 and 0° E) for the HadCM3, MPI-ESM, and iLOVECLIM simulations. Note: to capture variability, y-axis limits are not the same for each panel. Absolute surface*
*air temperature in the North Atlantic for the entire ensemble is shown in Fig. S3e – h.*

Likewise, *MPI_global_ice6gc* and *MPI_routed_ice6gc* both begin at ~4 °C at the start of the

deglaciation in the North Atlantic and then warm at a comparable rate, but slower than the HadCM3

simulations, warming ~3 °C by 15 ka BP rather than ~5 °C. The MPI simulations, like the HadCM3

simulations, also share a similar spatial pattern with an area of strong cooling in the Nordic Seas and

662 stronger warm patches off the coast of north-western North America and in the North Sea (Fig. S1).

This pattern appears to be independent of the ice sheet reconstruction, because *MPI_routed_glac* has

664 the same areas of relative cold and warmth at 18 ka BP, but the signal is weaker, likely because





*MPI_routed_glac* is ~5 °C warmer in the North Atlantic at the start of the deglaciation than the ICE-6G_C simulations. Temporally, however, *MPI_routed_glac* displays more surface air temperature variability in the North Atlantic with abrupt climate changes as large as 5 °C and AMOC decreases of ~9 Sv at ~18.2 and 15.2 ka BP, most likely following the higher-frequency variability in the meltwater input from the GLAC-1D ice sheet reconstruction (Fig. 2Fig. 6), but also because *MPI_routed_glac* is significantly colder at the LGM compared to its ICE-6G_C counterparts. Kapsch et al. (2022) show that the MPI simulations that are colder during the LGM lie closer to a critical threshold of AMOC variability. This aligns with the findings of Oka et al. (2012) and Klockmann et al. (2018) that demonstrate that the AMOC becomes more sensitive to perturbations, such as ice sheet topography, and the resultant wind stress, and $CO_2$ concentrations, when it is closer to an existing temperature threshold. Ensemble absolute surface air temperatures in the North Atlantic (Fig. S3e-h) show that multiple simulations in the ensemble are colder than *MPI_routed_glac* at the LGM, but only *MIROC's* AMOC appears to be close to a critical threshold of variability, as indicated by the changes in maximum AMOC strength towards 15 ka BP.

*iLOVE_routed_glac* has a similar, but less pronounced, variability of the AMOC and corresponding decreases in Greenland surface air temperature to *MPI_routed_glac* (Fig. 2b). However, in the North Atlantic, neither the *iLOVE_routed_glac* simulation nor *iLOVE_uniform_glac* exhibit significantly more variability than the ICE-6G_C iLOVECLIM simulations (relative to *MPI_routed_glac* and its ICE-6G_C counterparts). Spatially, the ICE-6G_C and GLAC-1D simulations are also nearly indiscernible (Fig. S1), except at the beginning of the deglaciation in the Southern Hemisphere, where surface air temperatures remain cooler for longer in the GLAC-1D simulations. This suggests that iLOVECLIM is less sensitive to freshwater perturbations than MPI-ESM-CR under these background conditions; however, this depends on how both modelling groups calculate their freshwater flux, which can vary, technically, even though the same ice sheet reconstruction is used (see section 3), as well as, and potentially more importantly, the fact that these simulations are performed with two very different models. For example, iLOVECLIM is an Earth system model of intermediate complexity (EMIC) with three atmospheric layers (see Table 1), whereas MPI-ESM-CR is an Earth system model (ESM) with 31 atmospheric levels, and thus can represent topographic feedbacks on the atmosphere with higher complexity and at finer-scale resolution. Unfortunately, more simulations using a GLAC-1D derived freshwater flux do not exist to compare to *MPI_routed_glac* and *iLOVE_routed_glac* and to get more robust results. Further simulations from other model types using both ice sheet reconstructions would be beneficial to understanding whether the systematic differences between the models are contributing to the differences in sensitivity to the freshwater forcing.





*4.4 Sensitivity of climate models to similar forcing(s)*

All simulations, with the exclusion of the UVic simulations, *TraCE-21ka,* and *FAMOUS,* use the

greenhouse gas forcing on the AICC20212 timescale, with an increase in atmospheric $CO_2$

concentration at ~17.5 ka BP. In contrast, in *TraCE-21ka* and *FAMOUS,* the $CO_2$ concentration does not

begin increasing until ~17 ka BP. This delayed increase in $CO_2$ postpones the warming of the

deglaciation in these simulations, as is evident in the tropical regions (Fig. 3Fig. 5, and S3). *MIROC*,

despite not having a delayed $CO_2$ increase, also displays delayed warming in the tropics, like that of

*FAMOUS*. This could be due to the stronger impact of the orbital forcing in *MIROC* taking precedent

over the $CO_2$ forcing earlier in the deglaciation (Obase and Abe-Ouchi 2019).

Contrasting sensitivities of the models used for the *TraCE-like* simulations is evident in the

response of the AMOC to the freshwater forcing and corresponding changes in Greenland surface air

temperature in the different models (Fig. 2). By 17 ka BP, all four simulations have reached a similar

and constant freshwater flux (with *iTraCE* ~0.05 Sv, or 33%, higher). The four simulations, however,

begin with a range of different AMOC strengths. *LOVECLIM* has the strongest LGM AMOC at ~28 Sv,

*TraCE-21ka* with the weakest LGM AMOC at ~12 Sv, and *HadCM3_TraCE* and *iTraCE* are in the middle

of the cluster, starting with an AMOC strength of ~24 Sv (see Table S1; Fig. 2g). Note that

*HadCM3_TraCE* and *iTraCE* start at 20 ka BP, whereas *LOVECLIM* and *TraCE-21ka* start at 21 and 22

ka BP respectively.

Despite beginning the deglaciation with the strongest AMOC, *LOVECLIM*'s ocean circulation is

also the most sensitive to the freshwater perturbation, causing its AMOC to crash to the weakest

AMOC state of all the simulations (Fig. 2g). The temperature change in the LOVECLIM simulation,

however, is comparable to the temperature change in *TraCE-21ka* despite the very different AMOC

responses to the freshwater forcing. The AMOC collapses to nearly 0 Sv, but Greenland surface air

temperature only decreases by ~5 °C.

The Greenland surface air temperature response in *HadCM3_TraCE* and *iTraCE* appears to be

impacted similarly by the change in AMOC strength, with both simulations following comparable

trajectories throughout the deglaciation despite *iTraCE* having a larger freshwater flux. Both

simulations exhibit an AMOC decrease of ~14 Sv and ~ -7 °C of temperature change between 19 and

16 ka BP. In addition, although *TraCE-21ka* and *HadCM3_TraCE* use the exact same freshwater flux,

the *HadCM3_TraCE* simulation exhibits a decrease in AMOC strength of over ~14 Sv and a

corresponding decrease in surface air temperature of ~10 °C in Greenland, whereas *TraCE-21ka*'s

AMOC strength weakens by only ~9 Sv and Greenland surface air temperature only decreases by ~4

730    °C. This suggests that the HadCM3 simulation is more sensitive to freshwater perturbations than





*TraCE-21ka*, but also that under the simulated climate conditions, Greenland surface air temperature
in HadCM3 is also more sensitive to corresponding AMOC changes compared to the other models.
Additional exploration would be interesting to determine what different aspects between
*HadCM3_TraCE* and *TraCE-21ka* could be contributing to the discrepancies in sensitivity (e.g.,
whether it could be the initial conditions, other boundary conditions, parameter choices, or simply
model structure). The lower sensitivity of CCSM3 to freshwater perturbations is further investigated
by He and Clark (2022) by rerunning *TraCE-21ka* but with no freshwater input during the Holocene.
This version of the simulation is in better agreement with proxy Holocene AMOC kinematic
reconstructions (McManus et al. 2004; Lippold et al. 2019).

The differences in model sensitivity are less observable in the simulations that apply
meltwater forcing in accordance with the PMIP4 protocol's ice sheet consistent recommendations, as
discussed in section 4.3.

*4.5 Meltwater paradox*

There has been ongoing debate on how much meltwater to input into simulations of the last
deglaciation, and these results highlight the impact of the decision. The debate has stemmed from a
so called 'meltwater paradox' that exists between the choice of large and geologically- inconsistent
meltwater forcings that successfully produce abrupt climate events versus glaciologically realistic
meltwater fluxes that do not. This paradox is particularly evident in the last deglaciation during
Heinrich Stadial 1 and the Bølling Warming. Heinrich Stadial 1, for instance, is associated with weak
ocean circulation strength (Lynch-Stieglitz 2017; Ng et al. 2018; Pöppelmeier et al. 2023a) and cold
climate conditions in multiple regions. There has been difficulty reconciling a weak AMOC concurrent
with a small amount of 'realistic' freshwater release, as determined by the ice sheet reconstructions,
in model simulations of the early deglaciation. Because of this, some model experiments (e.g.,
simulations in the *TraCE-like* meltwater group) have required, by design, overly-large quantities of
freshwater forcing to collapse their initially strong AMOCs and produce an abrupt cooling event like
that of the surface air temperature proxy records of the Hulu and Kulishu Caves (Wang et al. 2001;
758    Ma et al. 2012). Ivanovic et al. (2018) suggested that the AMOC weakening targeted in these
simulations is too large, and that a smaller meltwater flux inducing more modest North Atlantic
change may be sufficient to drive the recorded Heinrich Stadial climate. However, fully transient
simulations that include only meltwater that is consistent with the ice sheet reconstructions (i.e.,
*HadCM3_routed*, *MPI_routed_ice6gc*, *MPI_routed_glac*, *iLOVE_routed_ice6gc*, *iLOVE_routed_glac*, and





their corresponding *melt-uniform* simulations), do not achieve either the AMOC change nor the
surface climate signal of the Heinrich Stadial.

In this context, the MIROC last deglaciation simulation, is unique because it simulates the
weak AMOC and cold surface air temperatures of Heinrich Stadial 1 (Fig. 2h and S3h) and the
resumption of the AMOC of the Bølling Warming without releasing an unrealistically large amount of
freshwater (not shown as this paper only covers until 15 ka BP; see Obase and Abe-Ouchi 2019).
Instead, a cold, weak-AMOC state is achieved with a gradually increasing meltwater flux that remains
below the ice volume loss in the reconstruction. Notably, however, this simulation uses the *increasing*
meltwater flux to regulate the timing of the abrupt resumption of the AMOC and therefore, displays a
different sensitivity to freshwater input compared to the rest of the last deglaciation ensemble. This
is likely in part due to the very weak LGM AMOC state at the start of the simulation, which also plays
a role in the surface air temperature response and may make the simulation more susceptible to a
small freshwater flux. As the insolation and $CO_2$ concentrations increase, *MIROC*'s AMOC begins to
become unstable, and with the meltwater input into the North Atlantic gradually rising, it is
eventually pushed to a point where the AMOC suddenly strengthens at ~14.7 ka BP (Obase and Abe-
Ouchi, 2019; the precise timing of AMOC recovery depends on the meltwater forcing). The AMOC and
Greenland surface air temperature abruptly increase whilst the meltwater flux in the North Atlantic
is maintained. As the only PMIP4 last deglaciation simulation (LDv1 or previous) to simulate a weak
ocean circulation at the onset of the deglaciation and then its later rapid resumption *even* with a
continuous freshwater flux, this simulation may offer important insight to the conditions under which
abrupt deglacial climate change may occur. Nonetheless, even this model cannot reproduce the
Heinrich Stadial-Bølling Warming transition under Meltwater Pulse 1a-like freshwater forcing. Thus,
the meltwater paradox of the last deglaciation remains.

## 4. Conclusion

This study presents results from 17 simulations of the early part of the last deglaciation (20-15 ka
BP) performed with nine different climate models. Our analyses show the first assessment of these
simulations and display the similarities and differences between the model results as shown through
the timing of the deglaciation, spatial and temporal surface air temperature changes, the link between
the surface climate, ocean circulation, and $CO_2$ forcing, and how the different models respond to
different forcings. The impact of the chosen meltwater scenario on the model output has defined
every result of this multi-model intercomparison study. The course of the deglaciation is consistent
between simulations except when the freshwater forcing is above 0.1 Sv—at least 70% of the



simulations agree that there is warming by 15 ka BP everywhere excluding the location of meltwater

input. However, for simulations with freshwater forcings that exceed 0.1 Sv from 18 ka BP, warming
is delayed in the North Atlantic and surface air temperature correlations with AMOC strength are

much higher. The resultant impacts of $CO_2$ forcing and increasing insolation (i.e., ice sheet melt and
surface temperature warming) are reduced by the large freshwater fluxes imposed, delaying the

warming in the Northern Hemisphere for these simulations. Nonetheless, the average of the ensemble
displays the high latitudes beginning to deglaciate first in response to insolation and polar

amplification and later warming occurring in the tropics in correlation with the rising $CO_2$ trajectory.
The timing of the rise in $CO_2$ concentration differs between simulations depending on timescale of

the $CO_2$ reconstruction, delaying warming further in the tropics for simulations with a later $CO_2$
increase.

Simulations run by the same model (such as those from HadCM3, MPI-ESM, and iLOVECLIM)
show comparable surface climate patterns despite the use of a different ice sheet reconstruction or

the *melt-routed* versus *melt-uniform* freshwater scenarios. The main differences noted during this
time period include slower warming in the North Atlantic in the *melt-routed* simulations, additional

temporal variability in the GLAC-1D simulations, and faster warming in the GLAC-1D simulations.
Simulations run with different models, but similar boundary conditions, provide insight into the

sensitivity of the model to a particular forcing. We suggest that LOVECLIM's AMOC is the most
sensitive to freshwater perturbation and CCSM3's is the least sensitive; although, this is not

necessarily consistent with the sensitivity of the corresponding surface air temperature changes
because of complexity in how surface air temperature is linked to AMOC and other transient climate

forcings.

This multi-model intercomparison project has been beneficial to compare simulations of different

forcings to represent some of the uncertainty of the time period; however, it has also posed the
challenge of drawing direct model-to-model conclusions. It would be ideal to be able to compare more

simulations with the same experimental design to learn more about model sensitivities and test
additional plausible scenarios of climate changes during the last deglaciation. Hence, this study may

guide the design of future protocols for multi-model comparisons of the last deglaciation. One of these
protocols could also assist with narrowing down the uncertainties regarding the meltwater paradox;

for instance, the simulations that follow the *TraCE-like* meltwater scenario display larger variability
in the AMOC and Greenland surface air temperature, following more closely with proxy records of the

respective variables. However, to achieve this, the *TraCE-like* meltwater scenarios include freshwater
fluxes that are much larger than the amount deemed 'realistic' by the ice volume change in ice sheet



reconstructions of the time period. In contrast, simulations that follow the ice sheet reconstruction, show less agreement with the AMOC and Greenland surface air temperature proxy records, but show

a more gradual warming throughout the deglaciation that has more agreement with surface temperature proxy records, globally. Because meltwater input that is not realistic has such a large

impact on the results—dominating over other deglacial forcings, there is difficulty comparing simulations that do and do not choose this *TraCE-like* scenario. Thus, additional experiments with an

ensemble of models testing out the meltwater scenario options could assist with unravelling the current meltwater paradox.

## 5. Code availability

Python code will be posted on a Git Hub repository called '*pmip_ldv1_analysis_snoll*'.

## 6. Data availability

Data will be available through a University of Leeds DOI that is currently being set up.

## 7. Author contribution

The study conception was developed by the PMIP4 Working Group, consisting of RI, LM, TO, AA, NB, MK, UM, and PV. BS, LG, SS, and RI contributed to the study design, with LM, TO, and AA providing additional feedback and close communication with BS. The design of the experiments and running of them was performed by RI, LG, LM, TO, AA, NB, CH, FH, MK, UM, JM, and PV. Material preparation and data collection was performed by BS. The manuscript was prepared by BS with contributions from all co-authors, who read and approved the final manuscript.

## 8. Competing interests

LM is a member of the editorial board of Climate of the Past, but otherwise the authors declare that they have no conflict of interest. The authors consent to participation and publication.

## 9. Funding

BS is supported by the Leeds-York-Hull Natural Environment Research Council (NERC) Doctoral Training Partnership (DTP) Panorama under grant NE/S007458/1.

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
