# Peer review of "Title: A multi-model assessment of the early last deglaciation (PMIP4 LDv1): A"

_EGUsphere, 2023_

## Author Comment (AC4)

**Response to Reviewer #1**: Review of 'A multi-model assessment of the early last deglaciation (PMIP4 LDv1): A meltwater perspective' by Brooke Snoll et al.

Recommendation: Minor Revisions

This paper discusses a group of transient simulations of the early last deglaciation. It is a useful summary of the major features of the simulations. I have some minor comments on the paper.

*We thank the reviewer for their constructive and helpful comments which we have used to improve our manuscript. We address their points in blue and show changes made to the manuscript in* green.

The subtitle meltwater paradox is buried in the much more diverse discussions of other features of the simulations. Or it is only a very small part of the paper. Therefore, it should not be there. The value of this paper is mainly recording these features as a reference.

*We have changed the title to 'A multi-model assessment of the early last deglaciation (PMIP LDv1): A meltwater perspective'.*

"….whilst more recent modelling studies show a deep and strong ocean circulation ….."

This part of discussion should include a more recent proposal that, more likely, AMOC strength at the LGM is not too different from that at PI, in spite of the robust shallowing structure (Gu et al., 2020; Zhu et al, 2021).

*We moved this paragraph to the discussion as part of other suggestions to shorten the introduction and have a more thorough discussion. We have included the reviewer's input into this moved paragraph by changing it to say: 'There is debate on the strength of the LGM AMOC and how this initial state impacts the subsequent climate change of the deglaciation. Some observations have suggested a weaker and shallower LGM AMOC than present-day (e.g., Lynch-Stieglitz et al. 2007; Böhm et al. 2015; Lynch-Stieglitz 2017), with agreement from recent data-model comparison studies (e.g., Menviel et al. 2017; Muglia and Schmittner 2021; Wilmes et al. 2021; Pöppelmeier et al. 2023b). Whilst other ocean circulation proxy studies (e.g., McManus et al. 2004; Gherardi et al. 2005, 2009; Ivanovic et al. 2016; Ng et al. 2018) demonstrated a consensus of a vigorous but shallower AMOC coming out of the LGM (relative to the modern day) that subsequently weakened and shallowed (but remained active; Bradtmiller et al. 2014; Repschläger et al. 2021; Pöppelmeier et al. 2023b) during the abrupt transition to Heinrich Stadial 1. Recent modelling studies also have suggested between a deep and strong ocean circulation at the LGM (e.g., Menviel et al. 2011; He et al. 2021; Sherriff-Tadano and Klockmann 2021; Kapsch et al. 2022; Snoll et al. 2022) due to the presence of thick ice sheets (Oka et al. 2012; Sherriff-Tadano et al. 2018; Galbraith and de Lavergne 2019) and a shallow AMOC of similar strength to present-day (e.g., Gu et al. 2020; Zhu et al. 2021). '*

189: "….This creates a meltwater paradox, where the freshwater forcing required by models to produce recorded climate change is broadly in opposition to the meltwater history reconstructed from ice sheet and sea level records".

It should be pointed out that part of this seemingly inconsistent result for BA onset/M1A may be reconciled, partly, if the M1A meltwater flux is mostly injected into the Southern Ocean, as implied by
the reconstruction of sea level rise patterns.

We have added these sentences to address this comment: 'Meltwater Pulse 1a is a complex event
thought to be a culmination of contributions from the North American (Gregoire et al. 2012, 2016), Eurasian (Brendryen et al. 2020), and Antarctic (Weber et al. 2014; Golledge et al. 2014) ice sheets.
Whilst some studies have suggested that freshwater in the Southern Ocean could have contributed to the temperature changes seen in the North Atlantic during the Bølling Warming, recent studies (e.g.,
Ivanovic et al. 2018; Yeung et al. 2019) have demonstrated that the impact of meltwater pulses in the Southern Ocean on the climate are often restricted to the Southern Hemisphere, whereas North
Atlantic pulses have much farther-reaching and dominating affects. This creates a meltwater...'

215--: The reason most models do not produce the abrupt BA onset under a smooth meltwater forcing
is because the AMOC in those models are monostable. This stability, however, may be a bias, as discussed in previous works (e.g. Liu et al., 2014).

We reference Valdes (2011) earlier on in the introduction (paragraph 'One particularly challenging aspect...') making the same argument about models being too stable to be able to produce abrupt
events, we have added in the Liu et al., (2014) reference as well. We have also added in this sentence regarding the MIROC simulation to address this comment: '...the reconstruction. This study was able
to simulate spontaneous abrupt changes in AMOC thanks to multi-stability in their ocean circulation, as also seen in other modelling studies (Romé et al. 2022; Malmierca-Vallet et al. 2023).'

"4.1: timing of deglaciation" and Fig.2a-d: Greenland temperature. The smooth-looking Buizert curve at 19-18ka may be due to the muted d18O response by the expanding sea ice cover, instead of
declining temperature (He et al., 2021)

We have since updated Figure 2 with the new Buizert et al. (2018) Greenland temperature record
that has improved data constraints. Buizert et al. (2014) was reliant solely on d18O whereas the updated record also includes improved d15N at the onset of Heinrich Stadial 1. This record is less
smooth and shows some signs of a decrease in Greenland temperature around 17.5 ka BP, potentially corresponding with Heinrich Stadial 1, that the Buizert et al. (2014) didn't previously show.

626:" The AMOC for each of the HadCM3, MPI, and iLOVECLIM simulations is impacted by the chosen meltwater scenario during the deglaciation....". For this section of discussion, it may be useful to
diagnose the "effective" freshwater forcing on deep convection site, as discussed in He et al., 2020.

We agree that this would be an interesting analysis for future work. However, we do not have the
necessary ocean depth data to perform the analysis in the same way as He et al. (2020).

Fig.4, Fig.S4: Shakun et al 2012 is mostly SST, not surface air temperature!

We made this change. It should have said 'surface temperature'. The figure shows sea surface temperatures over water and surface air temperature over land.

Fig.8: caption. shouldn't "Spatial correlation..." be "Spatial distribution of the temporal correlation ..."?

We made this change for Fig, 7 and Fig. 8.

744: "4.5 Meltwater paradox…"…

The paradox is an obvious problem. Here, there is little discussion, or even speculation on the nature of the problem. Some discussions will be useful.

More specifically, it concerns with

      1. the AMOC response to climate forcing, and
2. the climate response to AMOC change.

    A climate model can be good for 2, but not for 1, and vice versa. So, it depends on the purpose of the
modeling work. If one is more interested in the response of global climate to AMOC change, the meltwater can be adjusted to make AMOC like reconstruction, and vice versa. One may speculate this
partly related to the model bias of the seemingly over-stable AMOC in current CGCMs. But, this problem seems to be present in EMICs too. If the AMOCs in these EMICs are bistable AMOC, this
paradox would involve additional factors.

    We agree that the inclusion of this into our discussion of the meltwater paradox would be useful. We
have added this paragraph to section 4.5 on the meltwater paradox: 'This renders the question of if our models have the right sensitivity to freshwater fluxes. There appears to be a consensus as to the
overall climate response to meltwater input in models and proxy records—the AMOC rapidly weakens, the North Atlantic cools, and sea ice forms, and the converse when meltwater input stops.
However, there is still less understanding and less agreement about how the AMOC responds to climate forcings. Because models appear to have AMOCs that are too stable, it is challenging to test
both the AMOC response to a climate forcing and the climate response to an AMOC change at the same time. If a modelling group is interested in the response of the global climate to changes in the AMOC,
they may be more inclined to adjust the meltwater pattern to trace the AMOC reconstruction, whereas if a modelling group is interested in the response of AMOC to a climate forcing, they may prefer to use
the meltwater derived from the ice sheet reconstruction. '

**Response to Reviewer #2**: Review of 'A multi-model assessment of the early last deglaciation (PMIP4 LDv1): A meltwater perspective' by Brooke Snoll et al.

The paper presents a study of a multi-model climate ensemble of the early last deglaciation. The
models differ in their forcings including the effect of greenhouse gasses and meltwater input. The models are stratified according to the implementation of the distribution of meltwater - as this seems
to be the dominating factor: melt-uniform, melt-routed, TraCe-like, and 'bespoke'. The paper discusses how the surface temperature and the AMOC develops in these models -- for example, the
warming begins at high latitudes while it is delayed in the tropics.

We thank the reviewer for their constructive and helpful comments which we have used to improve
our manuscript. We address their points in blue and show changes made to the manuscript in green.

I am not an expert in the field, but I find the paper interesting, although perhaps too long. In particular,
I think that the Introduction could be shortened (it is now 7 pages).

We agreed that the introduction was too long. We've removed much of the introduction of *TraCE-like*
simulations as many of these simulations are part of the MIP themselves. We removed additional details of some other simulations, such as the no-melt ones, and instead tried to summarize this in a
couple of sentences. We removed much of the background on PMIP and shortened it to just the background on the PMIP last deglaciation protocol. Lastly, we also removed the background on how
MIPs have been analysed in the past.

Specific comments:

Title: I had to look up 'reigns supreme' to get the exact meaning. Also, reading the paper I don't find a lot of results about the paradox, with section 4.5 being rather inconclusive. Perhaps it would be
better just to emphasize the large role of meltwater in the model results.

We have changed the title of the manuscript to 'A multi-model assessment of the early last
deglaciation (PMIP4 LDv1): A meltwater perspective'.

Figure 1: Panel c is not mentioned in the caption. Does the legend right of panel b also covers panel
c? What is the red curve in panel a? Perhaps you should not include references in the caption; it makes it hard to read.

We have decided to remove panel (c) as we did not actually reference it in the paper. We felt the references were important, however, to show readers where the records were coming from.

l398: I guess that what is important here (Fig. 4) is if the model mean is significantly different from 0. The measure used in the paper - 70 % agreement in sign - depends on the number of models and
corresponds here to 12 models of one sign and 5 of the opposite. It is not clear what the probability is that this will happen by chance.

The reviewer has raised an important point, and we had not previously checked the significance of these differences from the mean. We decided to perform this check. At each grid cell we have now
tested whether the mean was significantly greater or less than zero and if the individual simulation was greater than or less than zero at the same point using a 1 sample t-test with an alpha value of
0.05. If both were significantly different and in the same direction, than we counted that towards the number of simulations that agree with the sign of the mean in each location. The results of this analysis in comparison to the previous metric where we did not do the significance testing, were similar. There was still at least 70% of agreement with the sign of the mean throughout much of the Southern Hemisphere by 16 ka BP, and overall agreement increases through the deglaciation, but the regions surrounding the North Atlantic and the North Pacific take longer for the simulations to agree. We have updated the figure to show hatching with the significance testing.

l404: The subsection should be 3.1. The same for other subsections in section 3.

We made this change.

l467: I don't understand how the year of significant warming (Fig. 5) is determined. The description in the text should be improved. Are you, for each model, comparing 100 years centred about a t_0 with the 500 years in 20-19.5 ka BP, calculating the significance of this difference, and varying t_0 until p < 0.01? If this is the case, are you assuming all 100 years are independent? If they are not independent, the degrees of freedom in the t-test should be smaller and the differences will be less significant.

We have added a section called 'Analysis methods' that provides a more detailed description of how we performed this analysis. We have also realized, from other reviewer comments on this analysis, that it would be a more impactful analysis if it used an earlier reference period (i.e., 21 – 20.5 ka BP instead of 20 – 19.5 ka BP). We have since made this change in the main text and included the analysis with the later reference period in the supplementary information. Each sample are considered independent of each other, and each yearly cycle should be independent of each other.

The new section includes:

'One of the analyses used in this study was inspired by the year of first significant warming analysis performed by Roche et al. (2011). We define the first significant warming from the LGM using a statistical test. The LGM reference period is selected from the 500-year window between 21 and 20.5 ka BP for each simulation. Each of the simulations are then divided into 65 independent samples of 100 years between 20.5 and 13 ka BP for each grid cell. For each sample, we first performed a Fischer test on the variances of the reference and test samples to assess whether they differed or not. If the variances were equal, we performed a standard one-sided Student t-test with the alternative hypothesis as the sample period being warmer than the reference LGM period. If the variances were not equal, we performed a Welsch's test, or a t-test with two unequal variances with the same alternative hypothesis. The samples were tested at 99% confidence. If the sample was significantly warmer than the LGM reference period, then the grid point in Fig. 5 was assigned the central point of this sample. For example, if the 100-year sample between 16.2 and 16.1 ka BP at a specific grid point was determined to be significantly warmer than the reference period, then that grid point would be assigned the year 16.15 ka BP). This analysis excludes two of the simulations (*HadCM3_TraCE* and *iTraCE*) due to data availability before 20 ka BP. *LOVECLIM* was also not included due to a small drift between 21 and ~20.6 ka BP because of an adjustment in the ice sheet. This analysis was performed for all simulations with an earlier reference period (20 – 19.5 ka BP) and shown in the supplementary information. The remaining analyses in this study use a LGM definition of 20 to 19.5 ka BP to incorporate all simulations. '

Paragraph beginning at l520: I wonder if the comparison with the proxy data could be more detailed. It would be interesting also to see the comparison to individual models and not just the model mean.

We think that this would be a helpful and interesting addition to the manuscript, however, we also think that it would take a more detailed analysis that would be more impactful in a separate study. We are a part of a study that is doing this model-data comparison work on many of these same simulations. However, to address this comment, we've added the Shakun et al. (2012) temperature stack to the surface temperature maps in the supplementary information for each simulation (Fig. S1) and pointed to the previous model-data comparisons that some of the individual modelling groups have performed in their respective papers. We have added this short paragraph as some modelling groups have made previous data-model comparisons: 'For the comparison to some individual simulations, the Shakun et al. (2012) surface temperature stack is compared to surface temperature change from the LGM in Figure S1. Model-data comparison has also previously been performed by many of the individual modelling groups in their respective studies (see Table 1).'

l548: There seems to be something missing from the text here. Also, Fig. 6 is not discussed much in the text. For example, what is the reason for the peculiar shape of the curves in the TraCe-like experiments?

We agree that we neglected to discuss Fig. 6 to a helpful extent. We have addressed this comment by adding further discussion of Fig. 6 including why the *TraCE-like* simulations demonstrate this L-shaped curve. We also fixed the first sentence so that it was clearer to read. The text now reads:

'As $CO_2$ increases, surface air temperature increases, as demonstrated by the increasing trends on each panel of Fig. 6. Surface air temperature is also higher when the AMOC is stronger, clearly shown by *LOVECLIM*. The simulations with smaller AMOC variation have a clearer relationship with $CO_2$ concentration (see *melt-uniform* panel and all the *melt-routed* simulations except for *MPI_routed_glac*; Fig. 6). The *TraCE-like* simulations each have a strong L-shaped curve in the relationship between $CO_2$ concentration and surface air temperature. This is because the initial large decrease in North Atlantic surface air temperature, representing Heinrich Stadial 1, occurs as AMOC slows down whilst the $CO_2$ concentration is relatively constant (Fig. 1b). However, after ~18 ka BP (timing dependent on the $CO_2$ record used by the modelling group), $CO_2$ concentration begins increasing alongside a slow surface air temperature increase in each simulation.'

Figs 7 and 8: What are the units of the slopes? Are they Temp/AMOC or AMOC/Temp (Fig. 7)? Instead of looking at the influences of AMOC and CO2 on the temperature individually, the authors could try a multiple linear regression. As it is now, the analysis could be influenced by the correlations between CO2 and AMOC.

The units of the slopes are AMOC/Temp and CO2/Temp, so we use temperature as the dependent variable in these correlations. We updated the figures to add the units of the slopes.

We have previously run a multiple linear regression as well as a non-linear Random Forest regression with AMOC and $CO_2$ versus surface air temperature in Greenland and for global mean temperature. The results of this were like the individual regressions between AMOC and SAT and $CO_2$ and SAT but were more difficult to interpret. The simulations with larger changes in AMOC (i.e., the *TraCE-like* simulations and *MPI_routed_glac*) had stronger correlations between AMOC and Greenland SAT than the simulations with less AMOC changes. For the simulations with less AMOC changes, $CO_2$ has a stronger correlation with Greenland SAT. This was a bit less clear for the regression performed between AMOC, CO2, and global surface air temperature. The results of this regression showed a stronger correlation between $CO_2$ and global surface air temperature for each of the simulations.

Because of this discrepancy between locations and the difficulty in separating the individual influences of AMOC and $CO_2$ on surface air temperature, we decided to instead show the relationships
between each variable separately and spatially. This is to determine how the relationship between AMOC, $CO_2$, and SAT varies in space whilst also looking at the individual influences of the variables
on SAT. We do recognize that despite the separation of the two variables in this analysis, they are still influenced by each other. For example, in the $CO_2$ vs SAT figure, the *TraCE-like* simulations display a
negative correlation between $CO_2$ and SAT in the North Atlantic even though this relationship should be positive in all locations. This negative correlation is because during the period of 20 – 15 ka BP,
these simulations display a decrease of SAT despite an increase of $CO_2$ concentration. It is difficult to completely separate the influence of each individual variable on surface air temperature, even in a
multiple linear regression because of the collinearity between AMOC and $CO_2$ themselves, but we felt that this was an interesting and helpful way to discuss this relationship despite the caveats.

Examples from the random forest regression performed between AMOC, $CO_2$, and Greenland SAT:

[Figure]

l667: Can the large abrupt changes in MPI_Routed_glac (panel b in Fig 9) be related to the forcing in Fig. 1?

Yes, the figures referenced were supposed to be 1 and 2 instead of 2 and 6; we made this change. We do suspect that the large abrupt changes in *MPI_routed_glac* are partly due to the large variability in
the GLAC-1D freshwater forcing shown in Fig. 1. However, iLOVECLIM uses a similar freshwater forcing scheme for the routed, GLAC-1D simulation and does not display the same abrupt events.
Because of this, we also suspect that the MPI simulation is more sensitive to the meltwater input, as we have indicated in the next lines of the paragraph.

**Response to Reviewer #3**: Review of 'A multi-model assessment of the early last deglaciation (PMIP4 LDv1): A meltwater perspective' by Brooke Snoll et al.

Recommendation: Minor Comments

Summary

The author team investigates the early last deglaciation using a multi-model ensemble of 17 simulations. The authors take advantage form the fact that different modelling groups handle external forcings differently. Some even tested this handling using the same model. The authors identify that how meltwater is introduced to the North Atlantic strongly impacts the climate evolution of the deglaciation. The authors further found that the climate response to freshwater input is model dependent but also dependent on other forcings such as $CO_2$ and ice sheet configuration.

General

The paper presents a classical model intercomparison approach. It is overall well structured and well written, also though it is in some parts a bit lengthy. So, I encourage the author team to shorten the manuscript. Besides I have some minor to major recommendations prior to a possible publication in Climate of the Past.

We thank the reviewer for their constructive and helpful comments which we have used to improve our manuscript. We address their points in blue and show changes made to the manuscript in green.

Comments

Title: Given that the meltwater paradox is only weakly touched in the manuscript it gets too much weight being mentioned in the title and the abstract.

We have removed the 'meltwater paradox' from the title and the abstract, however, we recognize that much of the results of the paper do intertwine with discussion on meltwater input and how much is used for each modelling group. Because of this, we still think that we have a 'meltwater perspective' to this intercomparison and have included this in the title and abstract instead.

The abstract is rather long, please shorten it.

We have shortened the abstract and have tried to make it more concise and clearer to the points we want to touch on in the manuscript.

L52-55: I do not understand this sentence. Maybe this could be removed.

In the updated abstract, this sentence is removed.

L55-57: This is a rather general statement and I think can be removed.

In the updated abstract, this sentence is removed.

L87: please change to "suggested a weaker"

We have changed this sentence to 'Some observations have suggested a weaker and shallower AMOC than present-day during the LGM (e.g., Lynch-Stieglitz et al. 2007; Böhm et al. 2015; Lynch-Stieglitz 2017), with agreement from recent data-model comparison studies...'.

L88: please change to "studies showed a deep"

We made this change.

L92: please change to "showed"

We made this change.

L96: please change to "demonstrated"

We made this change.

L99-100: This is a rather long sentence, so I suggest splitting it here: "... Ng et al. 2018). Modelling studies ... suggested ..."

This paragraph has been moved to section 4.5 in effort to shorten the introduction, and the sentences have been edited based on other reviewer comments as well. We have shortened the sentence this
reviewer mentioned. The paragraph now reads: 'There is debate on the strength of the LGM AMOC
and how this initial state impacts the subsequent climate change of the deglaciation. Some
observations have suggested a weaker and shallower LGM AMOC than present-day (e.g., Lynch-Stieglitz et al. 2007; Böhm et al. 2015; Lynch-Stieglitz 2017), with agreement from recent data-model
comparison studies (e.g., Menviel et al. 2017; Muglia and Schmittner 2021; Wilmes et al. 2021; Pöppelmeier et al. 2023b). Whilst other ocean circulation proxy studies (e.g., McManus et al. 2004;
Gherardi et al. 2005, 2009; Ivanovic et al. 2016; Ng et al. 2018) demonstrated a consensus of a vigorous but shallower AMOC coming out of the LGM (relative to the modern day) that subsequently
weakened and shallowed (but remained active; Bradtmiller et al. 2014; Repschläger et al. 2021; Pöppelmeier et al. 2023b) during the abrupt transition to Heinrich Stadial 1. Recent modelling studies
also have suggested between a deep and strong ocean circulation at the LGM (e.g., Menviel et al. 2011; He et al. 2021; Sherriff-Tadano and Klockmann 2021; Kapsch et al. 2022; Snoll et al. 2022) due to the
presence of thick ice sheets (Oka et al. 2012; Sherriff-Tadano et al. 2018; Galbraith and de Lavergne 2019) and a shallow AMOC of similar strength to present-day (e.g., Gu et al. 2020; Zhu et al. 2021). '

L132: CCm2 is not explained.

We added the full name for CCSM3 in parentheses.

L143-...: I think there is also some papers discussing how freshwater is implemented to the North Atlantic and how this affects the response of the AMOC. For example, Stocker et al (2007) but also
references in there.

Stocker, T.F., A. Timmermann, M. Renold, O. Timm, 2007, Effects of salt compensation on the climate
model response in simulations of large changes of the Atlantic meridional overturning circulation, J. Climate 20, 5912-5928

We added '...that they enter the ocean (depth and latitude/longitude) and how they are implemented, as it determines the efficiency of convection disruption (e.g., Stocker et al. 2007; Roche et al. 2007,
2010; Smith and Gregory 2009; Otto-Bliesner and Brady 2010; Condron and Winsor 2012; Ivanovic et al. 2017; Romé et al. 2022)' to this sentence to incorporate this comment.

L151-156: The sentence is awkward, something is missing, and it is rather long, so please clarify.

       We split this sentence into two: 'The choice of a model's boundary conditions in the palaeo setting
(e.g., ice sheet geometry) can influence its sensitivity to freshwater perturbation. For example, Romé
       et al. (2022)'s simulations have an oscillating AMOC, whereas the simulations by Ivanovic et al. (2018)
do not, and Kapsch et al. (2022)'s demonstrated various climate responses in simulations of the last
       deglaciation with different ice sheets.'

L152: Please also check Yoshimori et al. 2010, which also show oscillations of the AMOC under cold
       conditions.

Yoshimori, M., M. Renold, C.C. Raible, T.F. Stocker, 2010, Simulated decadal oscillations of the Atlantic
       meridional overturning circulation in a cold climate state. Clim. Dyn. 34, 101-121

Thank you for sharing this reference with us; however, we feel that this work does not fit in this
       location. In this part of the paragraph, we are trying to reference how the choice in ice sheet geometry,
for example, can influence a model's sensitivity to freshwater perturbation. We feel that the
       comparison between Rome et al. (2022) and Ivanovic et al. (2018) as well as Kapsch et al. (2022)'s
work directory shows this. We otherwise don't really investigate oscillatory behaviour in this
       manuscript, but future work will benefit from looking at previous studies on oscillations.

L164: "have performed" to "performed" Please check in the entire manuscript the tense.

       We made this change.

L210: conducted

       We made this change.

L233: demonstrated

       We made this change.

L235-238: The sentence does not read good, please change.

       Whilst making changes to shorten the introduction of the manuscript, this sentence was placed in a
different location and shortened to: 'Whereas Gregoire et al. (2015) demonstrated that orbital forcing
       caused 50% of the reduction in North American ice volume, greenhouse gases caused 30%, and the
interaction between the two caused the remaining 20% in their couple climate-ice sheet simulations.'

       L239: Suggestion: "Sun et al. (2022) showed the effect of these forcings on the sensitivity of the AMOC
using multiple …"

       We made this change.

L247-279: I think these two paragraphs are not necessary. PMIP could be mentioned later, see next
       comment 19.

L282: Suggestion: "… simulations available from PIMIP4 (add references here) to better understand
       …"

370 We have removed most of the first paragraph on PMIP and the design of the PMIP4 last deglaciation protocol, whilst keeping a couple sentences as an introduction. We have also removed all the second

372 paragraph mentioned.

  L294: "The comparison is based on …"

374 We made this change.

  L452: What is meant by "This meltwater forcing presents itself in a higher variability"?

376 We changed 'presents itself' to 'is evident'.

  L536-539: This sentence is awkward, please clarify.

378 We have removed this sentence as its meaning was not clear based on this comment and author consensus.

380 L551: Which figure do you refer here, Fig 7 or Fig 8. My guess is Fig 7.

  Our figure references seemed to have gotten messed up, thank you for pointing this out. We have

382 addressed this issue. We meant to reference both figure 7 and 8 in this instance.

  L581: I suggest to also reference Fig 7 here, so: "… temperature (Fig. 7) for HadCM3_TraCE, …"

384 We made this change.

  L620: "In this study we include multiple …"

386 We made this change.

  L627: "see section 4.1). However, …"

388 We made this change.

  L670: showed

390 We made this change.

  L675: "Ensemble absolute surface air temperature in the North Atlantic" makes no sense so please

392 remove "Ensemble"

  We made this change.

394 L697: "the models contribute to the "

  We made this change.

396 L715: Missing comma before respectively.

  We made this change.

398 L739: No line break as the next paragraph is only one sentence.

  We made this change.

400 L752-758: The sentences are a bit strange, so please reformulate.

We have changed these sentences to: 'There has been difficulty reconciling a weak AMOC in model simulations of the early deglaciation with the small amount of 'realistic' freshwater release, as determined by the ice sheet reconstructions. Because of this, some model experiments (e.g., simulations in the *TraCE-like* meltwater group) have, by design, required overly-large quantities of freshwater forcing to collapse their initially strong AMOCs and produce an abrupt cooling event such as that shown by surface air temperature proxy records of the East Asia (e.g., Wang et al. 2001; Ma et al. 2012).'

L792: How can an "impact" define a "result"? This sentence needs revisions.

We have changed this sentence to: 'The impact of the chosen meltwater scenario on the model output is evident in each result of this multi-model intercomparison study.'

L798: Please change "resultant impact" to "impact".

We made this change.

L817: "... project compares simulations ..."

We made this change.

L818: "... however it poses the ..."

We made this change.

**Response to Reviewer #4**: Review of 'A multi-model assessment of the early last deglaciation
(PMIP4 LDv1): A meltwater perspective' by Brooke Snoll et al.

Recommendation: Major Comments

Summary:

This study presents a climate model intercomparison of the last deglaciation, which includes results
from nine climate models using a range of boundary conditions and forcings. The introduction
provides a thorough background on previous research linking freshwater fluxes, ice sheet geometry,
AMOC, and abrupt climate events that motivates the author's particular focus on the impact of the
quantity, timing, and distribution of freshwater forcings in models. The authors present a detailed
discussion of the ensemble from 20-15 ka BP and the results of their analyses, which is compelling
(except for one analysis) but would benefit from summaries in each section of the main findings. The
warming detection analysis from Roche et al. (2011), I believe, is incorrectly implemented. Overall,
this study is an important contribution and lays the groundwork for future intercomparison studies
aimed at resolving the deglacial meltwater paradox. The study would benefit, however, from more
details on how the results should inform future ensembles of simulations of the last deglaciation.

We thank the reviewer for their constructive and helpful comments which we have used to improve
our manuscript. We address their points in blue and show changes made to the manuscript in green.

Major Comments:

(1) The analysis in section 4.1 (lines 567-470 and Figure 5) to detect the start of the warming out of
the LGM, which is essentially a change detection analysis, incorrectly uses a baseline that overlaps
with the analysis period. The authors follow the method used in Roche et al. (2011) with the exception
that they use 20-19.5 ka BP as the reference LGM climate instead of an earlier time period or control
simulation. It is my understanding that the authors then conduct a one-sided Student's t-test between
this reference period and 100-year samples starting in 20 ka BP. The timing of the first 100-year
sample is not explicitly stated in the text, but can instead be inferred from Figure 5 which shows the
start of LGM warming occurring as early as 20 ka BP. It is the overlap of the analysis samples and the
reference period that can give an incorrect result for the start of the LGM warming. Here are a few
examples of this:

a) If the reference period has a cooling trend which is followed by a warming trend beginning in 19.5
ka BP, then this analysis may find that the warming started in the 20-19.9 ka BP sample simply
because this part of the reference period is the warmest.

b) If entire reference period (20-15 ka BP) has a warming trend, then this analysis will likely find that
the warming started near the end of the reference period or sometime afterwards. Such a result
should be accompanied by an explanation that this is the latest possible time the warming could have
started, and that it may have begun earlier.

c) The two examples above also show that this analysis will indicate that example (a) started warming before example (b) when in fact the opposite is true.

It may be that some of the model simulations do not suffer from these issues, but the fact that Figure 5 shows the start of warming at 20 ka BP for some locations and some models suggests that something akin to example (a) is happening. I cannot think of a scenario for which this analysis would correctly identify warming starting in 20 ka BP.

To correct this analysis, one option would be to search for the start of the warming beginning after the reference period (i.e., beginning with the 19.5-19.4 ka BP sample). As for example (b), this may mean that the start of the warming is detected after the real start. Such caveats and their implications should be stated clearly in the text.

We thank the reviewer for their detailed comment about this analysis. We determined that there was an error in the script we used to calculate the year of warming in which it was including the reference period in the samples. This has now been fixed, and we thank the reviewer for pointing this out. From their other comments as well as comments from other reviewers, we felt that we could further improve this analysis by starting with an earlier LGM reference period (21 – 20.5 ka BP instead of 20 – 19.5 ka BP) to capture more of the initial warming. This new analysis, however, did not include iTraCE and HadCM3_TraCE as we did not data before 20 ka BP. We have also removed LOVECLIM from this test as there is a small drift between 21 and ~20.6 ka BP due an adjustment in the ice sheet.

In the analysis with the earlier reference period, we noticed that there is significant warming earlier than 19.5 ka BP in most locations in all simulations (figure is shown below on the same colour scale and is now in the main text of the manuscript). The variance between the simulations decreases. Doing the analysis with the earlier reference period, however, excludes three of the four TraCE-like simulations, which we think skews the average and variance of the significant warming, as the TraCE-like simulations played a large role in the original discussion of the timing of the warming. We chose to show both analyses but include the analysis with the late reference period in the supplementary information.

We can thus say that there is significant warming in most simulations before 20 ka BP, but it is interesting to be able to see how additional warming is delayed by the freshwater fluxes in the TraCE-like simulations after 19.5 ka BP. We added an 'analysis methods' section to provide a space to better explain how this analysis was performed, and parts of the discussion were modified to match this edit.

[Figure]

(2) The Results/Discussion subsections provide a lot of interesting details, but the main findings in each section are not apparent. It would be helpful to have a summary of the main points at the end of
each section.

We have added short summaries to some of the sections to provide clarity on the results.

(3) In the Conclusion (lines 822-826), the authors mention a protocol to "assist with narrowing down the uncertainties regarding the meltwater paradox", but it's not clear from the rest of the paragraph
what this protocol would be. Since it seems to be a key contribution of this study to assist in designing such protocols, this deserves more discussion. At the end, the authors explain that additional experiments testing out meltwater scenarios would be beneficial. It would be helpful to know specifically how this study would inform the design of such experiments.

We have added this paragraph to the end of the conclusion: 'A protocol could assist with the design of additional experiments by outlining the use of different freshwater fluxes than modelling groups used previously. For the modelling groups that followed the PMIP4 meltwater scenarios, for example, it would be interesting to determine what 'trained' freshwater fluxes were required of their respective models to replicate the AMOC and Greenland proxy records as the *TraCE-like* groups and MIROC show, but also with different ice sheet reconstructions. This would teach us more about the sensitivity of each model to freshwater input and the impact of the ice sheet reconstruction on the AMOC's sensitivity. Similarly, if the *TraCE-like* groups performed simulations with more 'realistic' meltwater input, we would be able to compare to the previous PMIP4 meltwater experiments and narrow down the impact of different deglacial forcings on the climate trajectory throughout the deglaciation. This protocol would be beneficial to the understanding of the AMOC's sensitivity to freshwater fluxes as well as other climate forcings, such as $CO_2$ concentration and ice sheet configuration, and thus assisting with unravelling the current meltwater paradox. '

Minor Comments:

Line 49: Define the AMOC acronym.

We made this change.

Line 52: "demonstrate" --> "demonstrates"

This sentence was removed since the abstract has been restructured.

Line 86: "preceding" seems like the incorrect word here. Perhaps "subsequent"?

We made this change.

Line 91: "data assimilation modelling studies" is not an accurate portrayal of these studies. "data-model comparison studies" would be more accurate.

We made this change.

Line 103: It's not clear that any feedbacks are discussed in the rest of this paragraph.

We changed this sentence to: 'The AMOC pattern can be perturbed easily by changes in meltwater input into the North Atlantic.' We removed the mention of feedbacks.

Line 404: The section numbering is off for sections 4.1-4.5.

We made this change.

Line 434-443: In this paragraph the authors conclude that freshwater forcing is the dominant driver of the abrupt temperature changes in the HadCM3_TraCE and TraCE-21ka simulations. Though I

agree that this is likely the case, it is because simulations with different freshwater forcings yet similar other forcings and boundary conditions do not show this abrupt temperature decrease. This is the exact opposite reasoning that the authors use in this paragraph. The authors claim it is the fact that these two simulations are different in all respects except for the freshwater forcing that allows for the conclusion that freshwater forcing causes the temperature decrease. If many variables are different between the simulations, how can I attribute similarities or differences to any one aspect of the model?

Thank you for pointing this out. We have added this sentence to clarify the conclusion of this paragraph: 'Other simulations with similar boundary conditions to *HadCM3_TraCE* (i.e., *HadCM3_routed*) and *TraCE-21ka* (i.e., *FAMOUS*), but different freshwater forcings, do not show the large and abrupt decrease in the Greenland surface air temperature.'

Lines 551, 600, 669, and 703: Fix figure citations.

We made these changes.

Lines 562-564: I find this sentence to be misleading. Isn't it only the MPI model that shows this difference? The other option is the iLOVECLIM model, which doesn't show this difference. Please expand on why the MPI models show this difference.

The *iLOVE_routed_glac* simulation does have *higher* correlations between AMOC and surface air temperature than its ICE6-G_C counterpart in the southern hemisphere and a few areas in North

America and the North Atlantic. We tried to clarify this by adding in 'However, the *melt-routed* GLAC-1D simulations, in comparison to their ICE-6G_C same-model counterparts, exhibit higher correlations. The correlation between AMOC and surface air temperature in *MPI_routed_glac* increases in the Irminger and Nordic Seas from no correlation ($R^2$ is 0) in *MPI_routed_ice6gc* to an $R^2$

value of ~0.6. The slope of the MPI GLAC-1D simulation also changes from negatively correlated in most locations, to positively correlated. The differences between the iLOVECLIM GLAC-1D and ICE-

6G_C simulations are smaller. *iLOVE_routed_glac* does display higher $R^2$ values in the southern hemisphere and some locations in North America and south of Greenland; however, this correlation is still low (below 0.5). The larger differences in the MPI simulations could be due to the higher sensitivity of the model to the GLAC-1D freshwater flux than the ICE-6G_C freshwater flux, as described in more detail in section 4.3.'

We do touch on the fact that *MPI_routed_glac* lies closer to a critical threshold of AMOC variability than the other MPI simulations, as well as in comparison to the iLOVECLIM simulations in section 3.3. Kapsch et al. (2022) also go into more detail on this.

Lines 583-586: The impact of CO2 could be weakened or postponed, but it could also be that the CO2-caused warming is just as strong but is masked by the larger signal of the response to the freshwater forcing. Can this be ruled out by the results?

We have added these sentences to address this comment: '… The relationship between $CO_2$ and surface air temperature should be positive everywhere, so the negative correlation in the North Atlantic seen in the *TraCE-like* simulations suggests that the AMOC has a stronger influence on SAT

than $CO_2$ during the studied period (20 – 15 ka BP) for these simulations. Despite $CO_2$ concentrations beginning to increase, surface air temperatures are still decreasing…'

Line 619 & 624: This section discusses both climate and ice sheet forcings and boundary conditions. The section title and introduction to the section should reflect this.

We changed the title to 'Impact of different climate and ice sheet forcings and boundary conditions on model output'.

Line 629: Also reference a figure that shows the AMOC results, like Figure 2.

We made this change.

Line 705: By "stronger impact" do the authors mean the model may be more sensitive to the orbital forcing?

Yes, for better clarity, we changed this sentence to 'This could be due to the higher sensitivity of MIROC to orbital forcing , causing it to take precedent over the $CO_2$ forcing earlier in the deglaciation
(Obase and Abe-Ouchi 2019)'.

**Response to Reviewer #5**: Review of 'A multi-model assessment of the early last deglaciation (PMIP4 LDv1): A meltwater perspective' by Brooke Snoll et al.

Recommendation: Minor Comments

Summary

Snoll and co-authors have conducted and analysed a multi-model ensemble of simulations of part of the last deglaciation. They highlight areas of agreement and disagreement between the simulations,
focussing specifically on the impact of the use of different icesheet reconstructions and associated freshwater flux boundary conditions used in each. The comparison appears to have been conducted
carefully and is generally well presented, although the paper does feel over-long in places. I would recommend publications after consideration of minor issues listed below - and those raised by the
other reviewers, of course.

We thank the reviewer for their constructive and helpful comments which we have used to improve
our manuscript. We address their points in blue and show changes made to the manuscript in green.

General

I'm coming late to this paper and I can see it has a number of reviews already, so I'll try not repeat too much of what the others have already noted.

Complex climate model simulations of paleoclimate are a perennial topic and the results can often be very model and set-up dependent. This being the case I think the authors have done a good job in
pulling together the range of setups and results they're working with. The specific focus on the icesheet and freshwater boundary conditions makes for a good frame in my opinion and they've used
it to produce an analysis that is more than the sum of its parts and should be useful for other researchers in this area - even if the fundamental conclusion is that well-recognised issues and
internal inconsistencies between the boundary conditions and desired model behaviour are a major feature and that nothing here helps to resolve those.

Aside from some minor comments to follow, my major recommendation would be to tighten up areas that could be made more concise and to the point - the abstract and the Introduction in particular -
and to generally proofread for grammar (tense and number agreement mostly) and context (ie give dates as well as names when referring to paleo events) to aid the understanding of less-expert readers
who may not be familiar with the periods or PMIP conventions.

Specific Comments line 1: "PMIP LDv1" may be precise, but it's not helpful for those not already familiar with the topic -
so not a great candidate for inclusion in a title - and is never actually explained in the text. I'm not a fan of "The meltwater paradox reigns supreme" either - if this phrase's meaning and its implications were explained clearly in the abstract and made a major feature early in the Introduction then it could be justified, but as it is this is another rather confusing feature of the title for a non-expert.

We changed the sentence introducing PMIP to: 'To tackle such unknowns, the Paleoclimate Modelling Intercomparison Project phase 4 last deglaciation protocol version 1 (PMIP4 LDv1; Ivanovic et al.

2016) encompasses a broad range of models and is intentionally designed to be flexible.' – which now also includes a definition of PMIP4 LDv1. We have also changed the title to 'A multi-model assessment of the early last deglaciation (PMIP LDv1): A meltwater perspective'.

line 42 (and on): there's a lot of detail reported in this part of the abstract that I don't think helps a reader to see the key messages of the paper.

We have shortened the abstract to hopefully make it clearer and more concise.

line 64: what is the reference year for "Present" here? 1950, 2000?

The reference year is 1950. We have added this in.

line 72: "are" -> "were"

We made this change.

line 85 (and on): this is a very long Introduction and reads more like a general literature review of the field rather than focusing on previous findings of specific relevance.

We agreed that the introduction was too long. We've removed much of the introduction of *TraCE-like* simulations as many of these simulations are part of the MIP themselves. We removed additional details of some other simulations, such as the no-melt ones, and instead tried to summarize this in a couple of sentences. We removed much of the background on PMIP and shortened it to just the background on the PMIP last deglaciation protocol. Lastly, we also removed the background on how MIPs have been analysed in the past.

line 152: the clause in parentheses is very long and could be sentence or two in its own right if the result is worth saying.

We have changed this by splitting it into separate sentences. 'The choice of a model's boundary conditions in the palaeo setting (e.g., ice sheet geometry) can influence its sensitivity to freshwater perturbation. For example, Romé et al. (2022)'s simulations have an oscillating AMOC, whereas the simulations by Ivanovic et al. (2018) do not, and Kapsch et al. (2022) simulations of the last deglaciation test the climate response to different ice sheets.'

line 196: I don't think the author's italics are necessary.

We made this change.

Figure 1: The FAMOUS line in panel b is very unclear, if it is there at all? Is panel c actually referred to at any point in the text?

The FAMOUS $CO_2$ curve is very similar to the Joos and Spahni, 2008 curve. Because of the close resemblance, it is difficult to see. There are small discrepancies between ~19.8 and ~18.4 ka BP and around 15.7 ka BP. To clarify this, we've included this sentence '…thousand years later. The deglacial

$CO_2$ concentration for these two models is almost identical with some discrepancies between ~19.8 and 18.4 ka BP and about 15.7 ka BP. All…'

Panel c isn't referenced, thank you for pointing this out. We decided that this panel wasn't necessary to keep and have removed it from the figure.

line 366: Although often downplayed, it's not true to say that the UVic results are omitted from further discussion in this study is it?

We changed 'omitted from further' to 'omitted from parts of'.

line 478: "would be" should be "was"?

We made this change.

line 530 (and on): this phrasing might prompt a simplistic question that could be clarified: if the NAtl
is the region with most constraints and also the area with most variation across the simulations, why can we not just say that the simulation that's closest to the constraints must be right?

We added in the sentence: '…AMOC evolutions. It remains to be thoroughly tested if simulations that fit the constraints of the North Atlantic also fit the constraints of climate records from other locations.
The multi-model mean…'

line 551 (and elsewhere): the two "Fig" references are run into each other.

We made these changes.

line 554, 559: Two regression analyses have been done, temperature vs AMOC and temperature vs
$CO_2$. These sentences talk about $R^2$ values     without being very clear about which of these analyses they are refering to.

We have added in text to make this clearer.

line 669: Fig references run into each other

We made this change.

Figure 7: the HadCM3_trace and FAMOUS correlations have oddly patterned, very strong correlations
in places over Antarctica. What's going on there?

It seems like HadCM3 has some NaN values over Antarctica, as well as a little patch in FAMOUS. We
have checked this and made sure to keep these areas white. We have also changed the colour map for these figures so that white areas no longer look like high correlations but look like no correlation ($R^2$
of 0).

line 685: long sentence with confusing clause structure.

We split this sentence into: 'This suggests that, under these background conditions, iLOVECLIM is less sensitive to freshwater perturbations than MPI-ESM-CR. This is dependent, however, on how both
modelling groups calculate their freshwater flux, which can vary despite using the same ice sheet reconstruction (see section 3), as well as, and potentially more importantly, the fact that these
simulations are performed with two very different models.'

line 705: "is" -> "are"

We made this change.

line 750 (and on): a reminder of the dates for these events would be useful in this paragraph for non
experts.

We made this change for Heinrich Stadial 1 and the Bolling Warming.

line 765: this long summary of Obase and Abe-Ouchi (2019) seems to go into more detail than is
necessary to convey the relevant feature of their simulation.

We removed two of the sentences to make this summary more concise, but this discussion section
was also edited based on other reviewer comments.

line 768: why was this cut-off date chosen for this study? It would seem that including the Bolling
Warming and the end of Heinrich1 that occurs just beyond the cut-off would be a very useful feature
to discuss in a paper on the topic of deglacial AMOC and freshwater forcing.

After 15 ka BP, the models vary even more significantly than they do in the earlier part of the
deglaciation. It was determined that it would be best to go into detail on what is happening after 15
ka BP in a separate paper to this one. However, a recently submitted study to Climate Dynamics
(Takashi et al., In Review) looks at these simulations until 11 ka BP but for the Southern Hemisphere.

line 836: the Code and Data availability statements are clearly currently unverifiable.

We have started putting together the DOI, just waiting on a couple things, and the scripts are being
added to the Git Hub repository after some finishing touches as well.